# Eco-evolutionary dynamics of adapting pathogens and host immunity

Pierre Barrat-Charlaix[1,2,3], Richard A Neher[1,2]*

[1]Biozentrum, Universität Basel, Basel, Switzerland; [2]Swiss Institute of Bioinformatics, Basel, Switzerland; [3]DISAT, Politecnico di Torino, Torino, Italy

## eLife Assessment

This **important** study provides a new perspective on how human immunity shapes the antigenic evolution of pathogens. By combining theory and simulation the authors make a **compelling** case for the importance of eco-evolutionary interactions in population-level virus-host dynamics, which arise due to coupling between the dynamics of immune memories and viral variants. Although the work does not propose improved data-driven viral forecasting methods, it makes a conceptual contribution that advances the field's understanding of this problem's intrinsic difficulty.

## Abstract

As pathogens spread in a population of hosts, immunity is built up, and the pool of susceptible individuals are depleted. This generates selective pressure, to which many human RNA viruses, such as influenza virus or SARS-CoV-2, respond with rapid antigenic evolution and frequent emergence of immune evasive variants. However, the host's immune systems adapt, and older immune responses wane, such that escape variants only enjoy a growth advantage for a limited time. If variant growth dynamics and reshaping of host-immunity operate on comparable time scales, viral adaptation is determined by eco-evolutionary interactions that are not captured by models of rapid evolution in a fixed environment. Here, we use a Susceptible/Infected model to describe the interaction between an evolving viral population in a dynamic but immunologically diverse host population. We show that depending on strain cross-immunity, heterogeneity of the host population, and durability of immune responses, escape variants initially grow exponentially, but lose their growth advantage before reaching high frequencies. Their subsequent dynamics follows an anomalous random walk determined by future escape variants and results in variant trajectories that are unpredictable. This model can explain the apparent contradiction between the clearly adaptive nature of antigenic evolution and the quasi-neutral dynamics of high-frequency variants observed for influenza viruses.

**\*For correspondence:**
richard.neher@unibas.ch

## Introduction

Many human RNA viruses adapt rapidly to evade pre-existing immunity and re-infect humans multiple times over their lifetime. The most prominent examples of this evolution are influenza virus and SARS-CoV-2 (*Roemer et al., 2023*; *Petrova and Russell, 2018*), for which the changing virus population is surveilled in great detail and vaccines are updated regularly. To improve the match between the virus population and the vaccine, several groups are working on predictive models to anticipate the variants that dominate future viral populations (*Morris et al., 2018*; *Meijers et al., 2023*).

A common framework to model the rapid evolutionary dynamics of RNA viruses is to consider a population located away from the fitness optimum and with many accessible beneficial mutations (*Tsimring et al., 1996*). In this setting, clones compete to accumulate beneficial mutations as quickly as possible. In a process called selective sweep, successful variants expand and tend to be the ancestors of the future population while less successful mutants eventually disappear. The resulting fitness

distribution is a wave traveling along the fitness axis, the so-called *traveling fitness waves Rouzine et al., 2003*; *Desai and Fisher, 2007*; *Neher, 2013*. As the pathogen circulates, hosts develop immunity which leads to a 'deterioration of the environment' for the pathogen which approximately balances the increase in average fitness due to adaptation.

The traveling wave framework has been extensively used in this context as it allows for a straightforward ways to approach the prediction problem: each variant is assumed to have a fixed fitness relative to other variants, and inferring the fitness of all competing variants should allow prediction of the future composition of the population. Indeed, current methods typically infer the instantaneous growth advantage of a strain based on past and present circulation and then project this growth advantage forward in time *Luksza and Lässig, 2014*; *Neher et al., 2014*; *Huddleston et al., 2020*. While future mutations can reshuffle the relative fitness of lineages and thereby limit predictability, in these models a lineage that is most fit at any given time is most likely to dominate in the long run.

One short-coming of the traveling wave approach is the lack of explicit representation of the epidemiological dynamics and of the host's immunity. Indeed, fitness is only an effective parameters that summarize the complex interplay between viral antigenic properties and the hosts' immune systems. As such, it cannot explicitly describe important phenomena such as the build-up of host immunity to new variants, variant-specific immunity, or the interaction between strains through antigenic cross-reactivity. Taking the hosts' immunity and viral cross-immunity into account has the potential to strongly improve predictions *Meijers et al., 2023* or explain why prediction is difficult (*Barrat-Charlaix et al., 2021*).

The interaction between epidemiological dynamics and hosts' immunity are often modeled using generalizations of the Susceptible-Infected-Recovered model (SIR) to include multiple viral strains *Gupta et al., 1998*; *Gog and Grenfell, 2002*. In this setting, the natural question is that of the ultimate fate of the pathogen: will it go extinct, diversify to the point of speciation, or reach the so-called Red Queen State where new strains continuously replace old ones *Yan et al., 2019*; *Marchi et al., 2021*; *Chardès et al., 2023*; *Rouzine and Rozhnova, 2018*. To remain tractable, these studies typically approximate population immunity as a low-dimensional landscape in which the viral population evolves and ignores the complex heterogeneity in the immunity of different individuals. Furthermore, immunity is often assumed to be long-lived, and evolution of the pathogen in a stable low dimensional landscape gives rise to traveling waves.

Here, we study how novel variants of a virus shape the host population's immunity, which in turn changes their own growth dynamics. To do so, we use a multi-strain SIR model combining immune waning and heterogeneous immunity of the hosts. Such heterogeneity has been demonstrated for influenza virus in individuals of different ages *Lee et al., 2019*; *Welsh et al., 2023*. We show that this model generically leads to a situation where novel immune evasive variants emerge. In a homogeneous population of hosts, this leads to a succession of selective sweeps where novel variants compete against each other and replace previously circulating variants. However, in a heterogeneous population with a more rapid waning of immunity, initially growing variants lose their selective advantage before reaching fixation due to immunological adjustment of the host population. The phenomenology of our epidemiological model is reminiscent of ecological systems such as consumer-resource models, where adaptation by one species shifts the global equilibrium and distribution of other species but does not necessarily result in a selective sweep *Good et al., 2018*. In these systems, adaptation can usually not be modeled by a fixed fitness parameter for each strain but rather depends on the composition of the population *Tikhonov and Monasson, 2018*.

Strain dynamics in our model differ qualitatively from what is expected in the traveling wave scenario. While adaptive mutations are highly overrepresented in genetic diversity, they cease having a growth advantage when reaching intermediate frequencies, a process we call 'expiring fitness.' Once the fitness effect of a mutation has expired, its frequency randomly changes up or down as subsequent adaptive mutations occur on the same or on different genomic backgrounds.

This resemblance to neutral evolution could have important consequences for predictability of viral evolution. It is interesting to relate this to the recent observations that the evolution of influenza is not as predictable as one would expect from typical models *Huddleston et al., 2020*; *Barrat-Charlaix et al., 2021*. In particular, we observed in *Barrat-Charlaix et al., 2021* that the frequency trajectories of mutants of A/H3N2 influenza show features that are expected in neutral evolution but hard to explain in a traveling wave framework.

## Results

### Multi-strain SIR model

We describe the interaction of several viral strains and host immunity using a Susceptible/Infected compartmental model, similar to those used in *Gog and Grenfell, 2002*; *Yan et al., 2019*. In the most general form, the model describes $N$ variants of the virus labeled $a \in \{1 \ldots N\}$ circulating among $M$ groups of hosts with distinct exposure histories labeled $i \in \{1 \ldots M\}$ (immune groups). These groups could be different age cohorts or could be geographically separated. For each group $i$, we define compartments $I_i^a$ and $S_i^a$ as, respectively, the number of individuals of this group infected or susceptible to strain $a$. We assume that the total population of hosts is 1 so that we always have $0 \leq I_i^a, S_i^a \leq 1$, and values of $I_i^a$ and $S_i^a$ can be interpreted as fractions of the host population.

As with usual compartmental models, we assume that the dynamics are driven by the interaction of susceptible and infected hosts, leading to infections and gains of immunity. The rate at which hosts of group $i$ susceptible to variant $a$ are infected by $a$ is $\alpha S_i^a \sum_{j=1}^{M} C_{ij} I_j^a$. Here, $\alpha$ is an overall infection rate while $C_{ij}$ represents the probability of an encounter between individuals of groups $i$ and $j$. Thus, the above rate takes into account infections with strains $a$ caused by hosts of all groups. Considering that infected hosts recover at rate $\delta$, we can thus write the dynamics for $I_i^a$:

$$\dot{I}_i^a = \alpha S_i^a \sum_{j=1}^{M} C_{ij} I_j^a - \delta I_i^a. \tag{1}$$

When a host of group $i$ is infected by strain $b$, it gains immunity against the infecting strain $b$, but also to other strains $a$ with probability $0 \leq K_i^{ab} \leq 1$. Thus, $S_i^a$ decreases at a rate proportional to $K_i^{ab}$ and to the number of hosts infected by $b$ for every strain $b$. Since susceptibles to $a$ are depleted by infections from other strains, the dynamics of all strains are coupled. This coupling is determined by the matrices $\mathbf{K}_i$ of dimension $N \times N$, which in general differ between groups $i$ with different prior exposure history. Additionally, the waning of immunity at a rate $\gamma$ causes immune hosts to re-enter the susceptible compartment. We can now write the dynamics of $S_i^a$ as

$$\dot{S}_i^a = -\alpha \sum_{b=1}^{N} \sum_{j=1}^{M} S_i^a K_i^{ab} C_{ij} I_j^b + \gamma(1 - S_i^a), \tag{2}$$

where the first term accounts for immunity gains (or loss of susceptibility) due to infections or cross-immunity while the second represents immune waning. This model introduced by *Gog and Grenfell, 2002* assumes that immunity builds up through exposure and not only through infection. This explains that the change in $S_i^a$ is simply proportional to $S_i^a \cdot I_j^b$, regardless of the susceptibility of hosts to strain $b$. Alternative models that require infection for acquisition of immunity have qualitatively similar dynamics, but are mathematically more complex (Appendix 1.5). We also represented loss of susceptibility to $a$ due to exposure to $a$ using a trivial cross-immunity term $K_i^{aa} = 1$.

An important property of our model is that the probability of generating cross-immunity can differ between groups. The motivation is that strains $a$ and $b$ may be perceived as antigenically different by some immune systems, leading to a low $K_i^{ab}$, but as highly similar by others, leading to $K_i^{ab} \simeq 1$. Such a heterogeneous response by different immune systems has been observed experimentally in the case of influenza: in *Lee et al., 2019*; *Welsh et al., 2023* for instance, it was found that a given mutation in an influenza strain may allow escape from the antibodies of some individuals, i.e., low $K_i^{ab}$, while it had little effect on the serum of other individuals, i.e., high $K_i^{ab}$. Heterogeneous immune response could be caused by varying histories of strain exposure for different individuals, for instance, due to differences in age or geographical region. If immune groups correspond to age cohorts, mixing between groups is rapid, and we can simplify the connectivity between groups to $C_{ij} = 1/M$. If immune groups are shaped by geographic differences in exposure, the connectivity would be close to 1 on the diagonal ($1 - C_{ii} \ll 1$) while off-diagonal terms would be small ($C_{ij} \ll 1$ for $i \neq j$).

### Invasion of an adaptive variant

Hosts' immune heterogeneity and strain cross-immunity play two different roles in the model. The latter allows the model to reach a non-trivial equilibrium where multiple strains co-exist, while the former affects the convergence to the equilibrium.

To illustrate this, we design a simple scenario with only two strains: a wild-type and a variant. Accordingly, indices $(a, b)$ describing strains will take values $\{wt, v\}$. We consider that the two strains share the same infectivity rate $\alpha$, which amounts to say that they would have the same reproductive rate in a fully naive population. The case where the two strains differ in intrinsic fitness is explored in detail in Appendix 1.7. In brief, as long as the difference in intrinsic fitness is not too large compared to cross-immunity effects, the qualitative results given below hold, while larger intrinsic fitness differences lead to more classical dynamics like selective sweeps.

In the first version of this scenario, we will only consider one immune group, that is $M = 1$. We can thus skip the indices $i, j \in \{1 \dots M\}$, and we only have one cross-immunity matrix **K** that we parametrize as

$$\mathbf{K} = \begin{bmatrix} 1 & b \\ f & 1 \end{bmatrix}, \tag{3}$$

with $0 \leq b, f \leq 1$. $b$ quantifies the amount of 'backward'-immunity to the wild-type caused by the variant: a large $b$ means that it is likely that an infection by the variant causes immunity to the wild-type. Conversely, $f$ quantifies the 'forward'-immunity: infections by the wild-type causing immunity to the variant. If $f = b = 1$, the two strains are antigenically indistinguishable, and thus essentially identical for the model. Conversely, if $f = b = 0$, the two strains are completely distinct and do not interact through cross-immunity.

The dynamical equations now take a simplified form:

$$\dot{S}^a = -\alpha S^a \sum_{b \in \{wt, v\}} K^{ab} I^b + \gamma(1 - S^a),$$
$$\dot{I}^a = (\alpha S^a - \delta) I^a. \tag{4}$$

We can immediately derive the equilibrium state for this simplified case. We first define the reproductive number of strain $a$ as $R^a = \alpha S^a / \delta$. $I^a$ grows when $R^a > 1$ and declines when $R^a < 1$. The equilibrium susceptibility is, therefore, $S^a = \delta / \alpha$, such that $R^a = 1$. On the other hand, the equilibrium prevalence is determined by the inverse of the cross-immunity matrix **K**:

$$I_{eq}^a = \frac{\gamma}{\delta} \left( 1 - \frac{\delta}{\alpha} \right) \left( \mathbf{K}^{-1} \vec{1} \right)_a, \tag{5}$$

with $\vec{1}$ being the vector $[1; 1]$. The order of magnitude of the prevalence is given by the ratio of the rate of waning $\gamma(1 - \delta/\alpha)$ and the recovery rate $\delta$. In the following, we frequently use values $\alpha = 3$ and $\gamma = 5 \cdot 10^{-3}$ in units of inverse generations $\delta$, i.e., we set $\delta = 1$. At equilibrium, this corresponds to a fraction of $\sim 0.003$ of the host population being infected at any time. If generation time is a week, which is roughly the case for respiratory viruses such as influenza virus or SARS-CoV-2, the fraction of hosts infected in any year is $\sim 0.15$, which is of similar magnitude as empirical estimates for influenza (***Kucharski et al., 2018***).

It is also straightforward to compute the fraction of infections caused by the variant at equilibrium, thereafter referred to as the frequency of the variant. We find that this frequency is

$$\beta = \frac{1 - f}{(1 - b) + (1 - f)}. \tag{6}$$

In the case where $b = f$, the variant will ultimately settle at frequency 1/2. This includes the case where $b = f = 0$, where the two strains are completely independent and do not interact. On the contrary if $b \neq f$, the final frequency of the variant can in principle be anywhere between 0 and 1. For example if $b > f$, the variant is more likely to cause immunity to the wild-type than the wild-type is to cause immunity to the variant. In this case, $\beta > 1/2$ and the variant will be the dominant variant.

We are primarily interested in an 'invasion' scenario where only the wild-type is initially present in the population, that is $I^v = 0$ at $t < 0$. Cross-immunity with the resident strain reduces the fraction of hosts susceptible to the variant below one even though it has not circulated yet. But the number of susceptible hosts is always larger than the equilibrium value $\delta/\alpha$ unless $f = 1$, As a result, the growth rate of the variant is initially positive and given by

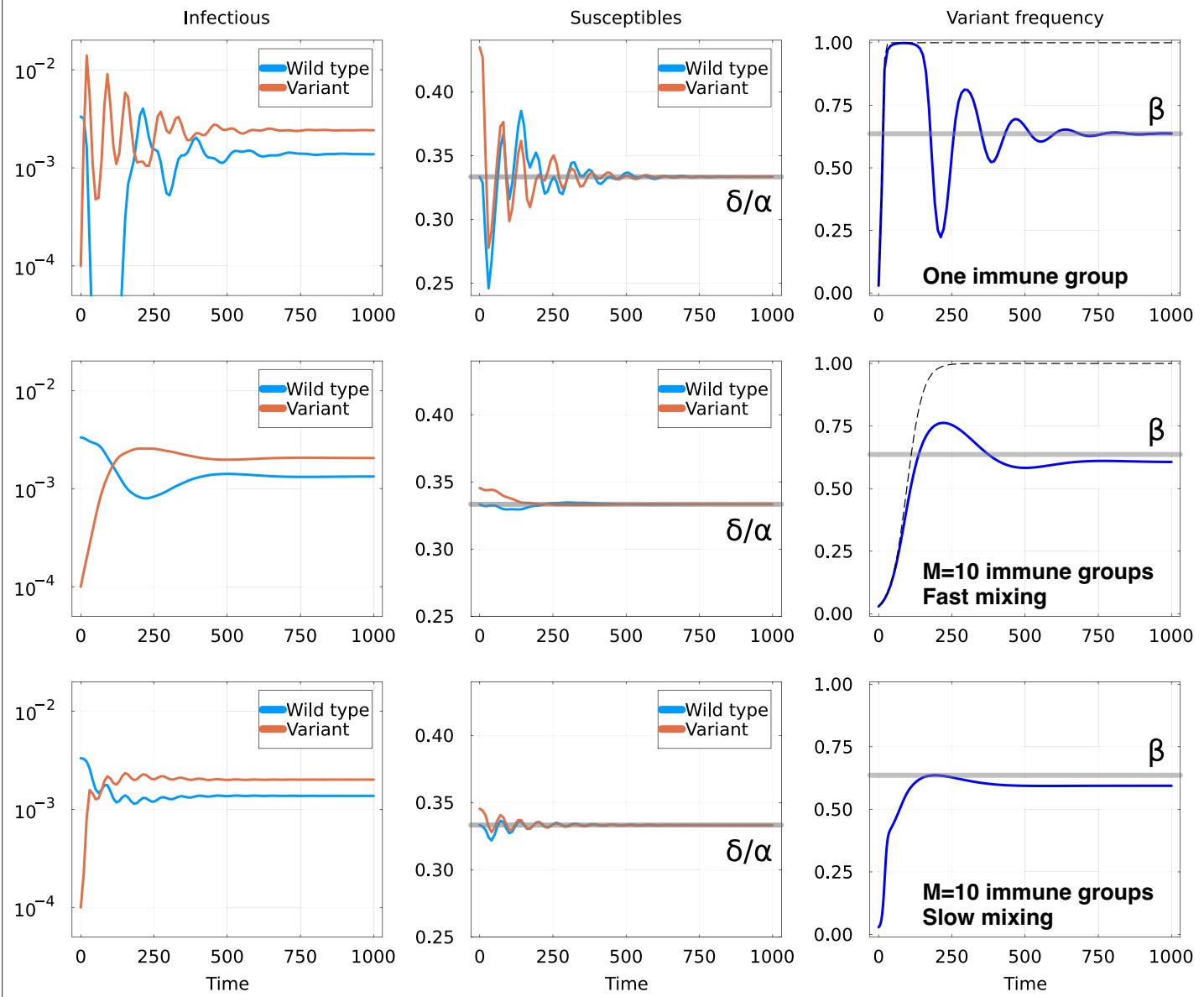

**Figure 1.** Invasion of an immune escape mutant. Top row: one immune group, Middle row: $M = 10$ immune groups and fast mixing $C_{ij} = 1/M$ and Bottom row $M = 10$ immune groups and slow mixing $C_{ij} = 1/10M$. Other parameters are the same for all rows: in units of $\delta$, we set $\alpha = 3$ and $\gamma = 5 \cdot 10^{-3}$, and $f = 0.65$, $b = 0.8$, $\varepsilon = 0.01$. For both rows, graphs represent: Left: number of hosts infectious with the wild-type and the variant; Middle: number of hosts susceptible to the wild-type and the variant, with the equilibrium value $\delta/\alpha$ as a gray line; Right: fraction of the infections due to the variant. The thick gray line shows the expected equilibrium frequency $\beta$ in the case with one immune group, given in **Equation 6**. The dashed line shows the trajectory of a constant fitness logistic growth with the same initial growth rate.

$$s(t = 0) = \frac{(1-f)(\alpha - \delta)}{\delta + f(\alpha - \delta)} \qquad (7)$$

The variant thus increases initially exponentially until it has become sufficiently frequent that it starts having a substantial effect on the immunity landscape, before eventually settling into an equilibrium with the wild-type. The details of the equilibrium reached by the system in the absence of additional mutant variants is given in Appendix 1.1. **Figure 1** explores different scenarios numerically.

The top row of **Figure 1** shows the dynamics of the model after the introduction of the variant in a homogeneous population ($M = 1$). As expected, the number of infections by the variant initially rises while the number of susceptibles $S^v$ decreases. However, as $S^v$ goes below the critical value $\delta/\alpha$, $I^v$

starts to decline and then oscillates around the equilibrium value before finally converging to it. The mathematical properties of these oscillations are discussed in Appendix 1.8.

However, these strong and slowly damped oscillations are not what is observed in circulating viruses. For instance, in the first oscillation in the specific example of *Figure 1*, the prevalence of the wild-type $I^{wt}$ goes down to microscopic levels and the frequency of the variant approaches one, see *Figure 1*. During stochastic circulation in a finite population of hosts the wild-type would likely be lost. The theoretical equilibrium that is reached at long times is thus not very relevant, and what would be actually observed in reality is a selective sweep by the variant.

Oscillations are the consequence of the rapid rise of the variant followed by an overshoot. This effect is mitigated by immunological heterogeneity, as shown in the following example with $M = 10$ groups. For group $i = 1$, the cross-immunity matrix $K_1$ takes the same form as in the previous scenario, given by *Equation 3*. However, for other groups, we assume that the two strains are virtually identical, with the cross-immunity having the form

$$K_{i>1} = \begin{bmatrix} 1 & 1 - \varepsilon \\ 1 - \varepsilon & 1 \end{bmatrix}, \tag{8}$$

where $\varepsilon \ll 1$. Our reasoning is that we expect an adaptive variant to escape the existing immunity for part of the host population, here immune group 1, while having little effect on the rest of the hosts.

One consequence of many groups that are indifferent to the variant is that globally the excess susceptibility to the variant is lower. If mixing is rapid, the initial growth rate of the variant is smaller by a factor of $M$ compared to the one-group case. If mixing is slow, the initial growth of the variant is as rapid as in the one-group case, but then spreads only slowly across groups. Globally, the frequency of the variant thus never reaches values close to one and population-wide oscillations are reduced.

The central and bottom rows of *Figure 1* show the result of the invasion for $M$ groups, respectively, for the rapid and slow mixing cases. In both scenarios, the initial number of hosts susceptible to the variant are now closer to $\delta/\alpha$. When mixing is fast, the frequency of the variant initially resembles a standard selective sweep (dashed line in *Figure 1*) before saturating, while dynamics are more complicated for the slow mixing case. Either way, the main effect of the immune groups is that the overshoot past the equilibrium is much smaller and the dampening of the oscillations stronger. As a result, the frequency of the variant approaches its equilibrium value without effectively sweeping to fixation before.

Notably, the equilibrium frequency in the above examples does not depend on $M$ and *Equation 6* remains valid for $\varepsilon = 0$. This invariance is a consequence of the fact that for $\varepsilon = 0$, the variant and wild-type strains are completely equivalent in immune groups $i > 1$ and equilibrium is only determined by cross-immunity in group $i = 1$ (Appendix 1.4). For small $\varepsilon$ the equilibrium shifts slightly, but *Equation 6* remains a good approximation.

While this simple two-strain model predicts that the two strains come to an equilibrium at frequency $\beta$, their frequency will of course continue to change due to the emergence of additional strains, which we will discuss below.

Even though the variant has a clear growth rate advantage when it appears, this does not result in it replacing the wild-type. This contrasts with the typical 'selective sweep' that occurs when the growth rate advantage stays constant, which is illustrated as a dashed line in the figure. We refer to frequency trajectories of a variant that at first rise exponentially before settling at an intermediate frequency as *partial sweeps*. As we will discuss below, such partial sweeps can lead to qualitatively different evolutionary dynamics and its predictability.

If the initial growth is due to higher susceptibility, it is misleading to think of it as an intrinsic fitness advantage of the variant. Instead, the initial growth is the result of an imbalance in the immune state of the host population, which gradually disappears as the variant becomes more frequent, as shown in the central panels of *Figure 1*. In this sense, our model is comparable to ecological systems where interaction between organisms cannot be fully explained using a fixed scalar fitness for each strain but rather depends on the composition of the viral and host population. In particular, the stalling of frequency increase gives rise to the partial sweep is reminiscent of consumer resource models *Tikhonov and Monasson, 2018*; *Good et al., 2018*, highlighting the link between ecological and epidemiological models. An important consequence of these dynamics is that predicting the

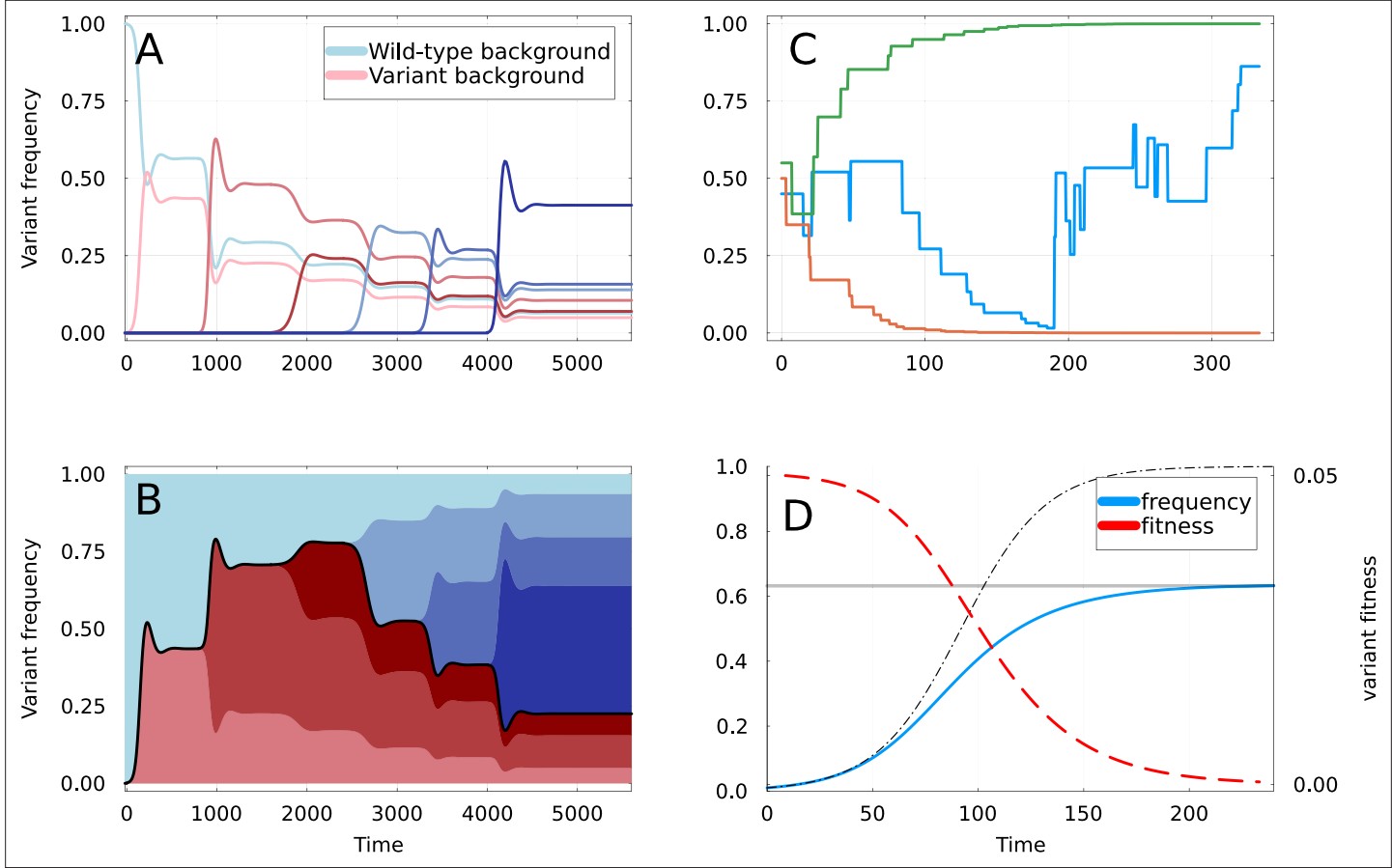

**Figure 2.** Dynamics of partial sweeps and subsequentfixation. (**A**) Simulation of Susceptible-Infected-Recovered (SIR) *Equation 1* & *Equation 2* with additional strains appearing at regular time intervals. The fraction of infections (frequency) caused by each strain is shown as a function of time. The first strain to appear at $t = 0$ is the variant of interest, and curves are shown in shades of red if they appear on the background of this variant, and of blue if they appear on the background of the wild-type. (**B**) Same as A but with frequencies stacked vertically. The black line delimiting the red and blue areas represents the frequency at which the mutations defining the original variant are found. (**C**) Three realizations of the random walk of *Equation 9*, all starting at $x \simeq 0.5$. Two instances converge rapidly to frequencies 0 and 1, corresponding to apparent selective sweeps, while the remaining one oscillates for a longer time. (**D**) Representation of a partial sweep using the expiring fitness parametrization of *Equation 11*. The frequency $x$ of the variant is shown as a blue line saturating at value $\beta$ (gray line). The thin dashed line shows a selective sweep with constant fitness advantage $s_0$. The fitness $s$ is a red dashed line, using the right-axis.

equilibrium frequency reached by the variant and its ultimate fate is hard from the observation of a partial frequency trajectory.

## Ultimate fate of the invading variant

In the invasion scenario discussed above, dynamics stop after the initial variant has reached an equilibrium frequency. However, as the viral population evolves, new adaptive mutants can appear. In the framework of the SIR model, a new strain translates into extending the cross-immunity matrix by one row and one column. Each new variant will perform its own partial sweep, and saturate at frequencies $\beta_2, \beta_3, \ldots$ sampled from some distribution $P_\beta$. This process is shown in panel A of *Figure 2*, using the SIR model to simulate up to $N = 7$ variants. For the sake of illustration, it shows a simple scenario where the initial variant appears at time 0 in a homogeneous wild-type population, and subsequent mutants appear at regular time intervals. Simulations are performed using $M = 10$ immune groups, resulting in a slight overshoot of the equilibrium frequency for each trajectory.

Here, we focus on the mutation or set of mutations $A$ that defines the initial variant. The initial growth rate advantage given by $A$ eventually disappears, meaning that after some times we can consider it as neutral. As subsequent mutants appear, they either do so on the background of the wild-type, in which case they do not carry $A$, or on the background of the initial variant in which case they

do carry $A$. If we suppose that recombination is negligible, the frequency of $A$ increases or decreases as each new variant undergoes its own partial sweep. This process is shown in panel B of *Figure 2*, with shades of red (resp. blue) indicating a variant carrying $A$ (resp. not carrying $A$). The thick line in between the red and blue surfaces indicate the frequency at which mutation $A$ is found, and in practice moves up and down randomly.

The scenario illustrated in *Figure 2* suggests that many aspects of the variant dynamics can be approximated by a simple abstraction: if $x$ is the frequency of a mutation $A$, a new variant has a probability $x$ to appear on the background of $A$ and thus carry $A$, and a probability $1 - x$ to not carry $A$. If new mutants emerge well separated in time with rate $\rho$, meaning that they reach equilibrium before the next variant emerges, and if new variants have a similar cross-immunity with all existing variants (see Appendix 1.6), the dynamics of $x(t)$ are described by a particular random walk: in each time interval $dt$, a partial sweep of amplitude $\beta$ occurs with probability $\rho dt \cdot P_\beta(\beta)$, changing $x$ in the following way:

$$x(t + dt) = x(t) + \Delta x, \text{ where } \Delta x = \begin{cases} -\beta x & \text{with probability } (1 - x), \\ \beta(1 - x) & \text{with probability } x. \end{cases} \tag{9}$$

For example, if a new mutant appearing in the background of $A$ does a partial sweep of amplitude $\beta$, the frequency of $A$ among the fraction of strains $(1 - \beta)$ not concerned by the sweep will still be $x$, and its frequency among the fraction $\beta$ of strains concerned by the sweep will be 1. Overall, this gives a frequency change of $\Delta x = (1 - x)\beta$. A similar reasoning gives us the frequency change when the new mutant appears on the wild-type background. Finally, if no sweep occurs in the time interval $dt$, that is with probability $1 - \rho dt$, $x$ remains unchanged. The resulting frequency dynamics of mutations have many similarities to the effect of 'genetic draft', that is the frequency dynamics of neutral mutations due to linked selective sweeps (*Gillespie, 2000*).

Examples of trajectories from the random walk are shown on panel C of *Figure 2*, all initially starting around $x_0 \simeq 1/2$. Two trajectories converge monotonically to 0 and 1. This is a consequence of one interesting property of *Equation 9*: the probability for $\Delta x$ to be positive increases with $x$, but the magnitude of the upward steps decreases as $1 - x$, and symmetrically with downward steps. This leads to trajectories leading almost exponentially to 0 and 1: it can in fact be shown that trajectories that *always* go downwards or upwards represent a finite and relatively large fraction of all possible trajectories (see Appendix 2.4). On the other hand, steps away from the closest boundary are unlikely but much larger, resulting in 'jack-pot' events (*Hallatschek, 2018*). This can be seen in the blue trajectory in *Figure 2*, which oscillates for a longer time.

It is also interesting to look at the moments of the step size $\Delta x$. The first two are easily computed, and we find

$$\begin{aligned} \langle \Delta x \rangle &= 0 \\ \langle \Delta x^2 \rangle &= \rho \langle \beta^2 \rangle_{P_\beta} x(1 - x). \end{aligned} \tag{10}$$

The first moment being 0 means that for the random walk, increasingly probable but small steps towards the closest boundary (0 or 1) are exactly compensated by rarer but larger steps away from the boundary. Importantly, this means that on average, the trajectory of mutation $A$ is not biased toward either fixation or loss, regardless of the frequency that the initial partial sweep brought it to. For instance, a mutation seen at frequency $x_0$ should on average stay at this frequency, which means in practice that in a finite population, it has a chance $x_0$ to reach 1 and fix, and a chance $1 - x_0$ to reach 0 and vanish.

On the other hand, the second moment resembles neutral drift *Kimura, 1964*: in neutral evolution, allele frequency also undergoes a zero-average random walk with the second moment having the form $x(1 - x)/N$ with $N$ being the population size. Therefore, this model would predict an 'effective population size' as $N_e^{-1} = \rho \langle \beta^2 \rangle_{P_\beta}$ completely independent of the size of the viral population. However, there are important differences to neutral drift: in neutral evolution, higher moments of order $k > 2$ decay as $N^{1-k}$ and are thus negligible in large populations, whereas here they are independent of $N$ and scale as $\langle \beta^k \rangle_{P_\beta}$. Depending on higher moments of $P_\beta$, allele dynamics will deviate qualitatively from neutral behavior.

## Abstraction as 'expiring' fitness advantage

In general, the dynamics of the SIR model proposed in *Equations 1&2* depend on the interactions between $N$ strains through an $N \times N$ cross-immunity matrix. While this model is useful to give a mechanistic explanation of partial sweeps, it is in general impractical to analyze and numerically simulate for many variants. The random walk model introduced above is simple to analyze and simulate, but assumes that variants rise to their equilibrium frequency instantaneously.

To explore the consequences of partial sweeps over broader parameter ranges, we propose an empirical model that has the same qualitative properties as the over-damped SIR, namely a growth rate that decreases as a strain becomes more frequent and partial sweep trajectories, but is simpler to analyze and simulate on a large scale. In this effective model, the growth rate $s$ of the variant is not explicitly set by the susceptibility dynamics in the host population, but instead decays at a rate proportional to the frequency $x$ of the variant:

$$\dot{x} = sx(1 - x) \quad \text{and} \quad \dot{s} = -xs. \tag{11}$$

The dynamic of $x$ in the first equation is simply given by the usual logistic growth with fitness $s$. To mimic increasing immunity against the invading variant, the growth advantage $s$ decreases proportionally to the abundance of the variant (second part of *Equation 11*). The initial value of $s_0$ is connected to the invasion rate of the SIR models given in *Equation 38*.

The dynamics of this new model are represented in panel D of *Figure 2*, with an initial frequency $x_0 \ll 1$ and an initial growth rate $s_0 = 0.05$. The initial growth of $x$ is identical to a classical selective sweep of fitness $s_0$ (represented as a dashed line). However, its fitness advantage gradually 'expires,' as shown by the red line in the figure. As the variant progressively 'runs out of steam,' its frequency finally saturates at a value $\beta$ given by (Appendix 2.2)

$$\beta = 1 - e^{-s_0/\nu}. \tag{12}$$

This final value $\beta$ depends only on the ratio between the initial fitness advantage $s_0$ and the rate of fitness decay $\nu$. For a large enough $s_0$, $\beta$ can be arbitrarily close to 1, meaning that this model still accommodates for full selective sweeps as a special case. In the general case, $x$ reaches its final value $\beta < 1$ and remains there forever unless other variants appear.

It is important to state that the main aim of this effective model is to qualitatively reproduce the phenomenology of the SIR, and in particular the partial sweeps, while being is easier to simulate. It recapitulates the salient feature of invading immune evasive variants: (i) initial exponential growth, and (ii) eventual saturation at an intermediate frequency. We can thus use it to analyze the long-term consequences of the random walk dynamics of *Figure 2*. However, we do not expect the frequency of the variant $x$ to have quantitatively equivalent dynamics in the two models. In particular, due to its simplicity, this model does not show the complex oscillatory behavior of the SIR model. Appendix 1.9 discusses in more detail the links between the parameters of the two models and the fundamental differences. While we can express the rate $\nu$ at which the growth rate declines in terms of the parameters of the simplest SIR models, for models with many groups or with oscillatory dynamics, the decay rate of the growth advantage should be interpreted as an effective parameter that captures a generic effect of reduced growth with increasing circulation.

## Consequences for predictability and population dynamics

Accurate prediction of dominant viral variants of the future could improve the choice of antigens in vaccines against rapidly evolving viruses. Specifically, if a potentially adaptive mutation is observed in a viral population, one would want to know if the corresponding variants will grow in frequency, and if yes to what point? The typical traveling wave framework would predict that fast-growing variants should keep on growing until an even fitter one appears. This way of thinking about the prediction problem has shown mixed results. In the case of A/H3N2 influenza, for instance, we showed that there are few signatures that suggest fit variants grow in frequency consistently (*Barrat-Charlaix et al., 2021*).

In *Figure 3*, we reproduce some of our results of *Barrat-Charlaix et al., 2021* and extend them to SARS-CoV-2. To quantify predictability, we ask the following question: given the state of a viral population at times $0, 1, \ldots, t$, what can we say about variant frequencies at times $t + 1, \ldots$? We performed a retrospective analysis of viral evolution and identified all amino-acid mutations that were observed

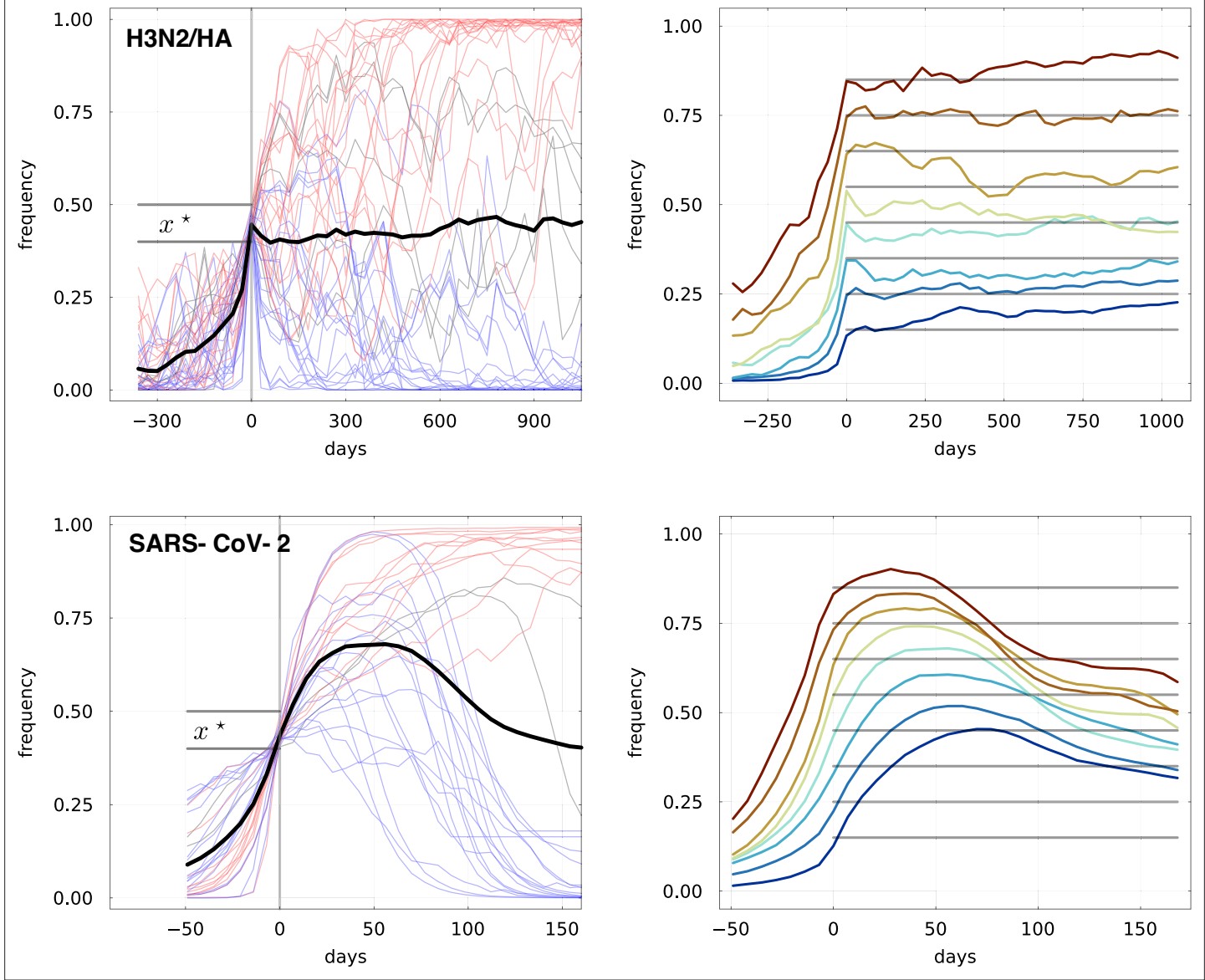

**Figure 3.** Retrospective analysis of predictability of viral evolution: frequency trajectories of all amino acid substitutions that are observed to rise from frequency 0 to $x^\star$ for Top: influenza virus A/H3N2 from 2000 to 2023, and Bottom: SARS-CoV-2 from 2020 to 2023. Left: all trajectories for $x^\star = 0.4$, with blue ones ultimately vanishing and red ones ultimately fixing. The average of all trajectories is shown as a thick black line. Right: showing only the average trajectories for different values of $x^\star$ (gray lines).

The online version of this article includes the following figure supplement(s) for figure 3:

**Figure supplement 1.** Example of mutation frequency trajectories that are increasing up to a frequency of 0.5 for H3N2/HA influenza and the expiring fitness model.

to grow from frequency 0 to an arbitrary threshold $x^\star$. Adaptive beneficial mutations should in principle be overrepresented in this group and if they provide a persistent fitness advantage, we would expect them on average to keep on growing beyond $x^\star$. *Figure 3* shows these trajectories for the amino acid substitutions in the HA protein of A/H3N2 influenza, using data from 2000–2023,, and the SARS-CoV-2 genome using data from 2020–2023. Panels on the left show all trajectories that reached $x^\star = 0.4$, with their average displayed in black. The panels on the right show the average trajectory for different threshold values $x^\star$ between 0.1 and 0.8.

While the dynamics of the variants of the two viruses can not be compared directly due to vastly different sampling intensities and different rates of adaptation, the qualitative patterns differ strikingly.

In the case of influenza, trajectories of seemingly adaptive mutations show little inertia and on average hover around $x^\star$ instead of growing. This surprising result is in line with the study in **Barrat-Charlaix et al., 2021** which used data from the period 2000–2018. On the other hand, trajectories of SARS-CoV-2 mutations show a much smoother behavior with steady growth beyond $x^\star$. On longer timescales, however, we observe a systematic decrease in frequency: this is explained by the particular initial dynamics of SARS-CoV-2, where new variants arose at a rapid pace and replaced old ones. This process is often called clonal interference and reduces long-term predictability.

In our setting of eco-evolutionary adaptation, the random walk model predicts that the probability of fixation of an immune evasive variant is given by the final frequency $\beta$ of its initial partial sweep. Subsequent allele dynamics and diversity are governed by an anomalous coalescent process driven by the random walk defined in **Equation 9**, leading to little predictability of evolution. This abstraction should hold when partial sweeps are instant and do not overlap, meaning that the rate $\rho$ at which new variants emerge is small compared to their initial growth rate $s_0$.

To explore the behavior of our partial sweep model in a more general setting, we simulate the evolutionary dynamics of a population under a Wright-Fisher model with expiring fitness dynamics. Simulations involve a population of $N$ viruses with a genotype where each position can be in one of two possible states $\sigma_i \in \{0, 1\}$. Fitness effects $s_i$ are associated with mutations at each position, and the total fitness of a virus is given by $F = \sum_i \sigma_i s_i$. At each generation, viruses with a fitness $F$ expand by a factor $e^F$, and the next generation is constructed by sampling $N$ individuals from the previous one. Following **Equation 11**, mutational effects $s_i$ decrease by an amount $\nu x_i \cdot s_i$, where $x_i$ is the frequency at which mutation $i$ is found in the population.

We simulate the emergence of adaptive variants in the following way. At a constant rate $\rho$, we pick one sequence position $i$ that has no polymorphism and set the fitness effect of the corresponding mutation to an initial value $s_i$, with an amplitude drawn from probability distribution $P_s$ and the sign chosen such that the mutation is adaptive. In practice, we use an exponential distribution $P_s \propto e^{-s/s_0}$, meaning that the typical magnitude of initial fitness effects are described by only one parameter $s_0$. The corresponding distribution of partial sweep size is described Appendix 2.3. At the same time, we introduce the corresponding mutant in the population at a low frequency, picking its background genotype from a random existing strain. The behavior of the model is determined by (i) the distribution $P_\beta$ of partial sweeps size depending on $\nu/s_0$, and (ii) the ratio of the variant emergence rate and their growth rate $\rho/s_0$, which determines how often sweeps overlap and interact. The probability of two sweeps overlapping is defined in Appendix 2, **Equation 42**.

We use this simulation to address the question of predictability: given the state of the population at generations $0, 1, \ldots, t$, can we predict its state at future times $t+1, \ldots$? Specifically, we ask whether we can predict the frequency $x(t + \Delta t)$ of a variant $A$, given it is at frequency $x$ at time $t$, as we did previously for the influenza virus **Barrat-Charlaix et al., 2021**, see **Figure 3**. The dynamics of isolated selective sweeps ($\rho/s_0 \ll 1$, $\nu/s_0 \ll 1$) should be perfectly predictable: after an initial stochastic phase when the variant is very rare, its frequency grows monotonically to fixation. This predictability decreases with increasing $\rho/s_0$ due to clonal interference (**Schiffels et al., 2011**; **Strelkowa and Lässig, 2012**), for example when an adaptive variant is outcompeted by an even more adaptive one. We also expects predictability to decrease with increasing $\nu/s_0$ since sweeps are then partial and their ultimate fixation is determined by subsequent variants with dynamics that resemble a random walk.

To quantify these effects, we select from a long simulation all rising frequency trajectories of adaptive mutations that cross an arbitrary threshold $x^*$. The results are shown in panel A of **Figure 4**, where we show the average $x(t)$ of rising frequency trajectories after crossing the threshold $x^* = 0.5$. We use three rates of fitness decay: $\nu \in [0, s_0/3, 3s_0]$ and low clonal interference $\rho/s_0 = 0.05$. The case $\nu = 0$ corresponds to a classical traveling wave scenario with constant fitness effects, and, as expected, is the most predictable: the average trajectory rises well above 0.5. For larger values of $\nu/s_0$, corresponding to a quicker decay of fitness, predictability gradually declines and becomes negligible for $\nu/s \gg 1$. Note that this matches quite well with the predictions from the random walk model where the average change in frequency $\langle \Delta x \rangle$ is null.

To explore parameter space more systematically, we quantify predictability as the probability of fixation $p_{fix}$ of rising variants that cross threshold $x^*$. In a perfectly predictable scenario with well-separated selective sweeps, $p_{fix}$ should be close to 1 regardless of $x^*$, while it should be equal to $x^*$ in an unpredictable setting such as neutral evolution.

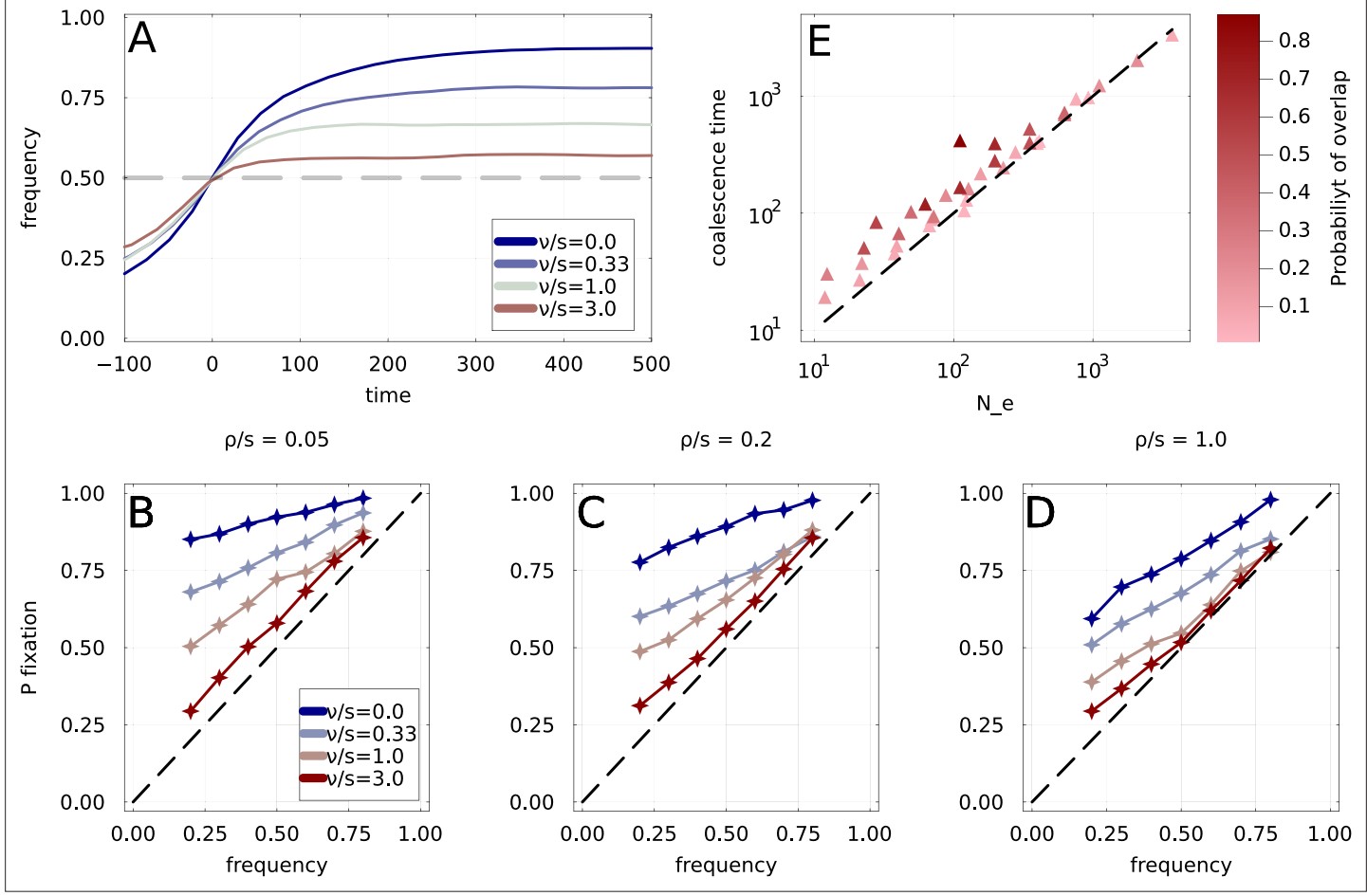

**Figure 4.** Simulations under the Wright-Fisher model with expiring fitness. (**A**) Average frequency dynamics of immune escape mutations that are found to cross the frequency threshold $x^* = 0.5$, for four different rates of fitness decay. If the growth advantage is lost rapidly (high $\nu/s_0$), the trajectories crossing $x^*$ have little inertia, while stable growth advantage (small $\nu/s_0$) leads to steadily increasing frequencies. (**B, C, D**) Ultimate probability of $p_{fix}(x)$ of trajectories found crossing frequency threshold $x$. Each panel corresponds to a different rate of emergence of immune escape variants, with four rates of fitness decay per panel. Increased clonal interference $\rho/s_0$ and fitness decay $\nu/s_0$ both result in a gradual loss of predictability. We use $s_0 = 0.03$. (**E**) Time to most recent common ancestor $T_{MRCA}$ for the simulated population, as a function of the prediction obtained using the random walk $N_e = 1/\rho\langle\beta^2\rangle$. Points correspond to different choices of parameters $\rho$ and $P_\beta$, and a darker color indicates a higher probability of overlap as computed in Appendix 2.2.

The online version of this article includes the following figure supplement(s) for figure 4:

**Figure supplement 1.** Probability of fixation of mutation Probability of fixation of mutations $p_{fix}(x)$ of mutation frequency trajectories found crossing the frequency threshold $x$.

In panels B, C, and D of **Figure 4**, the probability of fixations are shown for three values of $\rho/s_0$ and four values of $\nu/s_0$. Clonal interference increases when going from left to right among these panels (increasing $\rho/s_0$), while the intensity of fitness decay increases when going from blue to red curves (increasing $\nu$). Increasing either $\rho/s_0$ or $\nu/s_0$ reduces $p_{fix}$ towards the dashed diagonal corresponding to $p_{fix} = x^*$. However, as observed previously (**Barrat-Charlaix et al., 2021**), in the classic scenario with stable fitness effects $\nu/s_0 = 0$ considerable predictability remains even in cases of strong interference (blue curve in panel D and **Figure 4—figure supplement 1**). The strong interference setting is explored in more detail in Appendix 2.1 up to values $\rho/s_0 \simeq 30$, using similar simulations but without expiring fitness $\nu = 0$. **Figure 4—figure supplement 1** shows that even in these cases of strong interference, $p_{fix}$ remains significantly above the diagonal.

Finally, we use our simulation to investigate typical levels of diversity in the population and the time to the most recent common ancestor. One quantity that can easily be estimated from the random walk model is the average pairwise coalescence time $T_2$, that is the typical times separating two random

strains from their most recent common ancestor (MRCA). In Appendix 2.5, we show that under the random walk approximation $T_2 = 1/\rho \langle \beta^2 \rangle_{P_\beta}$, which in neutral models of evolution would correspond to the effective population size $N_e$. A more detailed analysis of the coalescent process reveals that the random walk approximation corresponds to the so-called $\Lambda$-coalescent *Schweinsberg, 2000*; *Berestycki, 2009*.

In panel E of *Figure 4*, the average time to the common ancestor of pairs of strains in the population is plotted as a function of $T_2$ predicted by the random walk model. Each point in the figure corresponds to one simulation of long duration with a given distribution of partial sweep size $P_\beta$ and a given $\rho$ setting $T_2$, with darker color indicating a higher probability of overlap as computed in Appendix 2.2. We find a good agreement between the empirical time to MRCA and the estimation from the random walk, at least as long as the probability of overlap between successive partial sweeps is small (indicated by shading). With increasing overlaps, coalescence slows down, and diversity increases: points in darker shades of red tend to have a larger time to MRCA than what is expected from the distribution of $\beta$. This is expected intuitively: if another adaptive variant emerges before the previous one has reached its final frequency, it has a lower probability of landing on the same background and thus tends to be in competition with the first variant. This leads to a smaller effective $\beta$ which slows the dynamics.

## Discussion

Evolutionary adaptation is often pictured as an optimization problem in a static environment. In many cases, however, this environment is changed by the presence of the evolving species, for example, because a host population develops immunity or a dynamic ecology. Here, we have explored the consequences of such eco-evolutionary dynamics in a case of host-pathogen co-evolution where different variants of a pathogen shape each other's environment through generation of cross-immunity.

Influenza virus evolution has been the subject of intense research with efforts to predict the composition of future viral populations (*Bush et al., 1999*; *Luksza and Lässig, 2014*; *Neher et al., 2014*; *Huddleston et al., 2020*). The A/H3N2 subtype in particular undergoes rapid antigenic change through frequent substitutions in prominent epitopes on its surface proteins (*Smith et al., 2004*; *Bhatt et al., 2011*; *Neher et al., 2016*; *Kistler and Bedford, 2023*). Given the clear signal of adaptive evolution, one might expect A/H3N2 to be predictable in the sense that variants that grow keep growing. Yet, it has been difficult to find convincing signals of fit, antigenically novel, variants that consistently grow and replace their competitors (*Barrat-Charlaix et al., 2021*; *Huddleston et al., 2020*). In contrast, SARS-CoV-2 evolution has been consistently predictable in the sense that dynamics are well modeled by exponentially growing variants that compete for a common pool of susceptible hosts. However, even in this case, taking into account the immune adaptation of hosts leads to a better description of variant dynamics *Meijers et al., 2023*.

We have shown that depending on (*i*) the heterogeneity of immunity in the population, (*ii*) the asymmetry between backward and forward cross-immune recognition, and (*iii*) waning or turn-over of immunity, the immune escape can either lead to dynamics dominated by selective sweeps, or to one were escape mutations have an initial growth advantage that dissipates before the variant fixes. The former scenario is observed when initial growth is fast, backward immunity high, and waning slow compared to variant dynamics. In this case, new variants can rise to high frequency driven by their own advantage and fix. Immunological heterogeneity slows down the initial rise, allowing for population immunity to respond and adjust before the variant has been fixed.

This process of partial sweeps reconciles two seemingly contradicting observations: HA evolution in human influenza A virus is clearly driven by adaptive immune escape and most substitutions are clustered in epitope regions (*Bhatt et al., 2013*). On the other hand, most substitutions does not sweep to fixation but tends to meander in a quasi-neutral fashion (*Barrat-Charlaix et al., 2021*). In the partial sweep scenario proposed here, diversity is dominated by immune escape mutations that are rapidly brought to macroscopic frequency by their initial growth advantage, but their ultimate fate is determined mostly by subsequent mutations.

In any real-world scenario, there will be a variety of mutations, including some mutations that perform complete selective sweeps, either because they escape immunity of a large fraction of the population ($M$ small), because they generate robust immunity against previous strains ('back-boost' *Fonville et al., 2014*), or because of the increase in the intrinsic transmissibility of the virus (for

example reverting a previous escape mutation that had a deleterious effect on transmissibility). The degree to which partial sweeps matter will vary from virus to virus and will change over time. Recently emerged viruses circulate in a homogeneous immune landscape and adapt to the new host for some time, consistent with rapid and complete sweeps of variants in SARS-CoV-2. Similarly, the influenza virus A/H1N1pdm, which emerged in humans in 2009, exhibited more consistent trajectory dynamics than A/H3N2 (*Barrat-Charlaix et al., 2021*).

More generally, qualitative features of the partial sweep dynamics investigated here are expected to exist in any system where the environment responds to evolutionary changes on time scales comparable to the time it takes for the adaptive variants to take over, leading to eco-evolutionary dynamics (*Pelletier et al., 2009*). In ecological systems involving eukaryotes, it is the evolutionary part of this interaction that is thought of as slow, while ecology is fast. In the cases of rapidly adapting RNA viruses in human populations with long-lived immunological memory, models often assume that viral adaptation is fast while hosts have long-lasting memory. The most complex and least predictable dynamics are expected when the evolutionary and ecological time scales are similar and different host-pathogen systems will fall on different points along this axis.

## Materials and methods
### Code availability

- Figures in the main text can be reproduced using a set of notebooks at https://github.com/PierreBarrat/ExpiringFitnessFigures (copy archived at *Barrat, 2024a*).
- Code for the simulations of the SIR model is available at https://github.com/PierreBarrat/PartialSweepSIR.jl (copy archived at *Barrat, 2024b*).
- Code for the population dynamics simulation is available at https://github.com/PierreBarrat/WrightFisher.jl (copy archived at *Barrat, 2024c*).
- Code to generate empirical frequency trajectories and their averages is available as scripts 'flu_fixation.py' and 'sc2_fixation.py' in https://github.com/nextstrain/flu_frequencies on branch 'fixation' (*Neher, 2024*).

### Data availability

Sequence data of influenza viruses was obtained from GISAID (*Shu and McCauley, 2017*). We thank the teams involved in sample collection, sequencing, and processing of these data for their contribution to global surveillance of influenza virus circulation. A table acknowledging all originating and submitting laboratories is provided as supplementary information.

Sequence data of SARS-CoV-2 viruses was obtained from NCBI and restricted to data from North America to ensure more homogeneous sampling. We are grateful to all teams involved in the collection and generation of these data for generously sharing these data openly.

## Acknowledgements

We gratefully acknowledge research support from the University of Basel (core funding) and the Swiss National Science Foundation (grant 310030_188547).

## Additional information

### Competing interests
Richard A Neher: Reviewing editor, *eLife*. The other author declares that no competing interests exist.

## Funding

| Funder | Grant reference number | Author |
|---|---|---|
| Schweizerischer Nationalfonds zur Förderung der Wissenschaftlichen Forschung | 310030_188547 | Pierre Barrat-Charlaix Richard A Neher |
| University of Basel | | Pierre Barrat-Charlaix Richard A Neher |

The funders had no role in study design, data collection and interpretation, or the decision to submit the work for publication.

### Author contributions
Pierre Barrat-Charlaix, Conceptualization, Formal analysis, Investigation, Visualization, Methodology, Writing - original draft, Writing - review and editing; Richard A Neher, Conceptualization, Supervision, Visualization, Writing - original draft, Writing - review and editing

### Author ORCIDs
Pierre Barrat-Charlaix ⓘ https://orcid.org/0000-0002-3816-3724
Richard A Neher ⓘ https://orcid.org/0000-0003-2525-1407

Reviewer #1 (Public review): https://doi.org/10.7554/eLife.97350.3.sa1
Reviewer #3 (Public review): https://doi.org/10.7554/eLife.97350.3.sa2
Author response https://doi.org/10.7554/eLife.97350.3.sa3

## Additional files

### Supplementary files
Supplementary file 1. GISAID acknowledgements table listing submitting and originating laboratories for the all sequences used in this study.

MDAR checklist

### Data availability
Accession numbers for all sequences from GISAID are provided as *Supplementary file 1*.

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

## Appendix 1

### SIR model

#### Equilibrium of the SIR model with one immune group

To help us compute the equilibrium reached by the SIR model, we introduce additional notation: the vectors $\vec{S} = [S^1, \ldots, S^N]$ and $\vec{I} = [I^1, \ldots, I^N]$ will, respectively, represent the compartments of hosts susceptible and infectious to each strain; the matrix $\mathbf{K}$ describes the cross-immunity; the vector $\vec{1}$ is a vector of dimension $N$ whose elements are all equal to 1.

We derive the equilibrium state for *Equations 1&2*. For each strain, $a$, equilibrium for $I^a$ is reached when $S^a = \delta/\alpha$. We thus have $\vec{S}_{eq} = \delta/\alpha \vec{1}$. Introducing this in the equation for the derivative of $S^a$, we obtain the following equilibrium:

$$
\begin{aligned}
\vec{S}_{eq} &= \frac{\delta}{\alpha} \vec{1}, \\
\vec{I}_{eq} &= \frac{\gamma}{\delta} \left( 1 - \frac{\delta}{\alpha} \right) \mathbf{K}^{-1} \vec{1}.
\end{aligned}
\tag{13}
$$

An interesting remark is that at equilibrium, $I$ is of order $\gamma/\delta \ll 1$. Note that the structure of $\mathbf{K}$ makes it invertible in most cases. Indeed, we impose $K^{aa} = 1$ and $0 \le K^{ab} < 1$ for $a \ne b$.

#### Equilibrium for two viruses with one immune group

We consider the case where two viruses are present, called wild-type (*wt*) and mutant (*m*). The cross-immunity is represented by a $2 \times 2$ matrix

$$
\mathbf{K} = \begin{bmatrix} 1 & b \\ f & 1 \end{bmatrix},
\tag{14}
$$

As shown in the previous section, the equilibrium is given by

$$
\begin{aligned}
S^{wt} &= S^m = \frac{\delta}{\alpha} \\
\vec{I} &= \frac{\gamma}{\delta} \left( 1 - \frac{\delta}{\alpha} \right) \mathbf{K} \vec{1},
\end{aligned}
\tag{15}
$$

where $\vec{I}$ stands for $[I^{wt}, I^m]$ and $\vec{1}$ for $[1, 1]$. It is straightforward to invert the cross-immunity matrix, and we obtain

$$
\begin{aligned}
I^{wt} &= \frac{\gamma(1 - \delta\alpha^{-1})}{\delta(1 - bf)}(1 - b) \\
I^m &= \frac{\gamma(1 - \delta\alpha^{-1})}{\delta(1 - bf)}(1 - f).
\end{aligned}
\tag{16}
$$

Note that without cross-immunity, the number of infected by either virus would be $\frac{\gamma(1 - \delta\alpha^{-1})}{\delta}$. Positive values of $b$ and $f$ thus have the effect of lowering the equilibrium values of $I^m$ and $I^{wt}$ with respect to the absence of cross-immunity.

It is interesting to compute the fraction of infections due to the mutant at equilibrium. This is easily derived from the relations above:

$$
\frac{I^m}{I^{wt} + I^m} = \frac{1 - f}{2 - b - f}.
\tag{17}
$$

A few observations can be made:
- if $b = 1$ and $f < 1$, then the wild-type vanishes at equilibrium and the mutant reaches frequency 1. In this case, the presence of the mutant alone is enough to keep $S^{wt}$ to its threshold value $R_0^{-1}$, making it impossible for the wild-type to grow.
- Inversely, if $f = 1$, then the mutant stays at frequency 0.
- If $b, f < 1$, the mutant will reach a finite frequency $x$, with $x > 0.5$ if $b > f$ and $x < 0.5$ if $b < f$.

## Equilibrium without the mutant

We first derive the equilibrium situation before the mutant virus is introduced in the case with only one immune group. We remind that in this case there is only one cross-immunity matrix which has the form

$$K = \begin{pmatrix} 1 & b \\ f & 1 \end{pmatrix},$$

where $b$ is the immunity to the wild-type caused by an infection with the mutant, and $f$ the reverse.

Since the mutant is absent from the host population, we assume $I^m = 0$. The equilibrium values for $S^{wt}$ and $I^{wt}$ are easily obtained from the dynamical equations:

$$S^{wt} = \delta/\alpha = R_0^{-1}$$
$$I^{wt} = \frac{\gamma}{\delta}(1 - S^{wt}). \tag{18}$$

We then set the derivative of $S^m$ to 0:

$$-\alpha f S^m I^{wt} + \gamma(1 - S^m) = 0$$
$$\rightarrow \quad S^m = \frac{1}{1 + f(R_0 - 1)} \tag{19}$$
$$= \frac{\delta}{\delta + f(\alpha - \delta)} > \frac{\delta}{\alpha} = S^{wt}.$$

Since we assume $f < 1$, the initial number of susceptibles to the mutant will be larger than $\delta/\alpha$, allowing the initial growth of the mutant. Using the dynamical equation for $I^m$, the initial growth rate of the mutant can be written as

$$\dot{I}^m(t=0) = \alpha S^m - \delta = \delta \left( \frac{\alpha}{\delta + f(\alpha - \delta)} - 1 \right). \tag{20}$$

If $f = 0$, the growth rate is $\alpha - \delta$, i.e., the one expected in a fully naive population. If $f = 1$ however, the growth rate is 0 as the wild-type confers perfect immunity to the mutant.

The equations above generalize to more immune groups. Cross-immunity matrices $\mathbf{K}_i$ now depend on parameters $f_i$ and $b_i$, and the initial number of susceptibles in immune group $i$ is given by

$$S_i^m = \frac{\delta}{\delta + f_i(\alpha - \delta)}. \tag{21}$$

In a given immune group $i$, the mutant growth rate is proportional to $S_i^m - \delta/\alpha$. The growth rate of the mutant will thus be initially faster in immune groups for which it is antigenically different, i.e., $f_i < 1$, than in groups where it is similar to the wild-type, i.e., $f_i \simeq 1$.

In the case of a well-mixed population, that is $C_{ij} = 1/M$, we can write the growth of the infections by the mutant $I^m = \sum_i I_i^m$ as an exponential growth with a time-dependent rate. In this case, the overall growth rate is given by the derivative of $I^m = \sum_i I_i^m$:

$$\dot{I}^m = \left( \frac{\alpha}{M} \sum_{i=1}^{M} S_i^a - \delta \right) I^m \tag{22}$$

In particular, using the invasion scenario from the main text with $\varepsilon = 0$ (i.e. $f_i = 1$) in $M - 1$ group and an arbitrary value $f$ in group 1, we obtain the following growth rate at $t = 0$:

$$\dot{I}^m = \frac{\delta}{M} \left( \frac{\alpha}{\delta + f(\alpha - \delta)} - 1 \right) I^m. \tag{23}$$

That is, the initial growth rate for $M$ groups is $M$ times smaller than the one in the single group case.

In the case of a non-well-mixed population, i.e., arbitrary $C_{ij}$, it is not possible to write a pseudo-exponential growth rate as in *Equation 22*. However, it is clear that the initial growth rate will also be smaller than in the single group case since the mutant initially only grows in group $i = 1$.

## Equilibrium with $M$ immune groups

For $M$ immune groups and arbitrary cross-immunity matrices $\mathbf{K}_i$, the equilibrium frequency of the two strains is not easy to compute. However, it is possible to give an analytical expression in the regime of fast mixing $C_{ij} = M^{-1}$ and when the two strains differ immunologically for only one group, i.e., for matrices.

$$\mathbf{K}_1 = \begin{pmatrix} 1 & b \\ f & 1 \end{pmatrix} \quad \text{and} \quad \mathbf{K}_j = \begin{pmatrix} 1 & 1 \\ 1 & 1 \end{pmatrix} \quad \text{for } j > 1.$$

Note that this corresponds to the situation studied in the main text with fast mixing and $\varepsilon \to 0$. We show here that in this case, the equilibrium frequency for all immune groups is the same as the one obtained for only one immune group with matrix $\mathbf{K}_1$. In other words, the expression for $\beta$ in *Equation 6* is still valid.

To prove this, we assume the following form for the solution of the dynamical equations:

$$\forall i \in \{1 \dots M\}, \forall a, \ S_i^a = M \frac{\delta}{\alpha} \frac{I_i^a}{I^a} \quad \text{and} \quad \frac{I_i^a}{I_i} = \frac{I^a}{I} = \nu^a, \tag{24}$$

where the index $a$ runs over all strains (here wild-type and mutant), and where we have defined the infectious levels for group $i$, for strain $a$ and globally:

$$I_i = \sum_a I_i^a, \ I^a = \sum_i I_i^a, \ I = \sum_{i,a} I_i^a.$$

Note that the second equation in Equation S24 means that the frequency $\nu^q$ of strain $a$ is the same across all immune groups and consequently also globally.

We now show that injecting these expressions of $S$ and $I$ in the dynamical system and solving for $\nu^a$ gives the expected result. First, note that with this choice of $S_i^a$, the derivative of $I_i^a$ given by equation *Equation 1* immediately vanishes. We thus concentrate on $\dot{S}_i^a$ given by *Equation 2*. For any immune group $i$ and strain $a$, we have

$$\dot{S}_i^a = 0 = -\delta \frac{I_i^a}{I^a} \sum_b K_i^{ab} I^b + \gamma \left(1 - M \frac{\delta}{\alpha} \frac{I_i^a}{I^a}\right).$$

where we have used $C_{ij} = 1/M$ and $I^b = \sum_j I_j^b$ to remove the sum on immune groups. Multiplying this equation by $\nu^a = I^a/I$, we obtain

$$-\delta I_i^a \sum_b K_i^{ab} \nu^b - \gamma \left(\nu^a - M \frac{\delta}{\alpha} \frac{I_i^a}{I}\right) = 0.$$

We now eliminate $I_i^a$ by using the expression $I_i^a = I_i \nu^a$:

$$\delta I_i \nu^a \sum_b K_i^{ab} \nu^b - \gamma \left(1 - M \frac{\delta}{\alpha} \frac{I_i}{I}\right) \nu^a = 0,$$
$$\delta I_i \sum_b K_i^{ab} \nu^b - \gamma \left(1 - M \frac{\delta}{\alpha} \frac{I_i}{I}\right) = 0.$$

Note that this last expression is true for all strains $a$. Considering any two strains $a$ and $b$, we can thus write

$$\delta I_i \sum_c K_i^{ac} \nu^c - \gamma \left( 1 - M \frac{\delta}{\alpha} \frac{I_i}{I} \right) = 0,$$

$$\delta I_i \sum_c K_i^{bc} \nu^c - \gamma \left( 1 - M \frac{\delta}{\alpha} \frac{I_i}{I} \right) = 0,$$

$$\Rightarrow \forall a,b \quad \sum_c (K_i^{ac} - K_i^{bc}) \nu^c = 0,$$

where the last expression is obtained by subtracting the two previous ones. First, we see that for $i > 1$ any frequency vector $\nu$ is a solution since $K_i^{ab} = 1$ for all $a, b$. For $i = 1$ and defining $\nu^m = \beta$ and $\nu^{wt} = 1 - \beta$ and using the expression for $\mathbf{K}_1$, we obtain

$$(1 - f)(1 - \beta) + (b - 1)\beta = 0$$

$$\Rightarrow \beta = \frac{1 - f}{(1 - b) + (1 - f)}$$

as claimed.

This result is not completely trivial and should be commented. In this setting, the mutant escapes immunity built by the wild-type for a fraction $1/M$ of the population, and yet it reaches the same frequency as in the case with one immune group. This can be rationalized as follows: for immune groups $i > 1$, the cross-immunity matrix is such that the wild-type and mutant strains are completely equivalent. If immune group 1 was not here, the mutant could thus equilibrate at any frequency between 0 and 1. Since it was initially introduced at a very low frequency, it would remain marginal in immune groups $i > 1$. However, since its 'natural' equilibrium frequency in group $i = 1$ is $\beta$ and since the groups are connected, equilibrium is reached when the mutant reaches frequency $\beta$ in all groups.

Note that if we take the situation of the main text with

$$\mathbf{K}_i = \begin{pmatrix} 1 & \varepsilon \\ \varepsilon & 1 \end{pmatrix}$$

and $\varepsilon > 0$, the expressions above do not hold. However, if $\varepsilon \ll 1$, the perturbation is small, and we expect an equilibrium frequency close to β, which is the case in *Figure 1*.

### Realistic modeling of the host's immune state

The SIR model proposed in the main text relies on the assumption of immunity acquisition through exposure. This explains terms like $-\alpha \sum_b S^a K^{ab} I^b$ in the derivative of $S^a$: acquiring immunity to $a$ through cross-immunity $K^{ab}$ requires a combination of prior susceptibility and exposure to strain $b$. Importantly, this does not depend on the immune state of the hosts with respect to strain $b$.

A more realistic representation would be one where acquiring immunity to strain $a$ from exposure to $b$ requires being *infected* by $b$. However, this would require keeping track of more precisely of the immune state of hosts, as we would need to separate hosts into two groups, namely

- hosts who are susceptible to both $a$ and $b$, and can thus acquire immunity to $a$ through infection by $b$;
- hosts who are susceptible to $a$ but immune to $b$, and can no longer acquire immunity to $a$ through infection by $b$.

To test whether our results are robust to such changes in hypothesis, we write a simple SIR model with two strains $a$ and $b$ where cross-immunity is only activated through infection rather than exposure. To properly track the immune status of the hosts, we introduce the groups $R^a$ and $R^b$, respectively representing hosts immune to only $a$ or only $b$, and $R^{ab}$ representing hosts immune to both $a$ and $b$. The compartment $R^0 = 1 - R^a - R^b - R^{ab}$ groups hosts susceptible to both strains. It is simpler in this case to write the dynamics in terms of compartments $I$ and $R$, rather than $I$ and $S$ as in the main text. For simplicity, we do not use immune groups here. The dynamics involve two equations for the infected:

$$I^a = \alpha(1 - R^a - R^{ab})I^a - \delta I^a,$$
$$I^b = \alpha(1 - R^b - R^{ab})I^b - \delta I^b,$$

(25)

and three for the immune:

$$R^a = \alpha(1 - K^{ba})R^0 I^a - \gamma R^a,$$
$$R^b = \alpha(1 - K^{ab})R^0 I^b - \gamma R^b,$$
$$R^{ab} = \alpha R^0 (K^{ba}I^a + K^{ab}I^b) + \alpha(R^a I^b + R^b I^a) - \gamma R^{ab}.$$

(26)

In *Appendix 1—figure 1*, we show both that the dynamical and equilibrium properties of this model are qualitatively the same as the one from the main text. On the left panel, we show that the dynamics of this new model do not differ qualitatively from the model of the main text. In particular, in the invasion scenario, the frequency of the variant converges to some equilibrium value after some oscillations. On the right, we show that this equilibrium value $\beta$ is different but relatively close to the one from the main text.

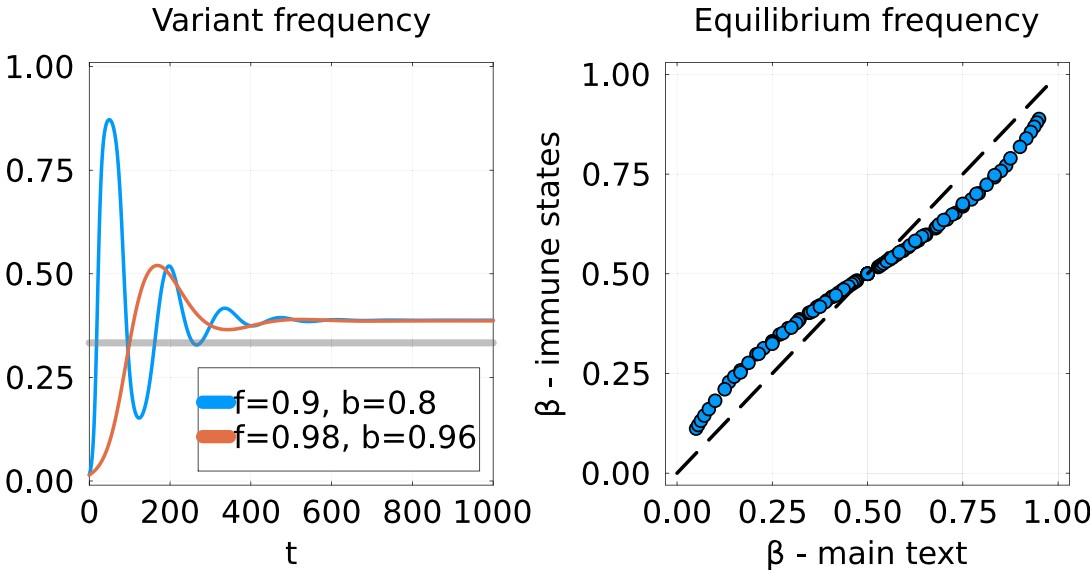

**Appendix 1—figure 1.** Comparison of SIR model implementations. Left: Dynamics of the frequency of the variant for the Susceptible-Infected-Recovered (SIR) model from *Equations 25; 26* using the invasion scenario from the main text. Two $2 \times 2$ cross-immunity matrices are used, with off-diagonal parameters $f$ and $b$ chosen to give the same equilibrium. The gray line represents the equilibrium that would be obtained using the model of the main text. Right: Equilibrium frequency $\beta$ for this new SIR model ($y$-axis) versus the $\beta$ from the main text ($x$-axis). Each point corresponds to a given pair $(f, b)$.

## Change in frequency when adding subsequent strains

This section shows that under certain condition adding a new variant to the SIR model does not change the relative frequencies of previous variants. This is an important condition for the random walk of the main text to be valid.

Here is a quick summary of the results proved below. Adding a variant to the SIR model involves adding a column $\vec{b}$ and a row $\vec{f}$ to the cross-immunity matrix, which can be given by two vectors. If these vectors only depend on one parameter, i.e., $\vec{b} = b \cdot \vec{1}$ and $\vec{f} = f \cdot \vec{1}$, then the relative frequency of previous strains is unchanged in the new equilibrium. What this means in practice is that the new strain must be at an equal 'antigenic distance' from all previous strains. A possible interpretation is an antigenic space of infinite dimensions: all mutations explore an antigenic region which is new.

We start from an initial situation where there are $N$ variants with an $N \times N$ cross-immunity matrix $\mathbf{K}$. At equilibrium, the number of hosts infected by each virus $a \in \{1 \dots N\}$ is given by the elements of the vector $I$ that can be computed from the cross-immunity matrix and parameters of the model:

$$I = \frac{\gamma(1 - \delta/\alpha)}{\gamma + \delta} K^{-1} \vec{1}_N, \tag{27}$$

where $\vec{1}_N$ is a vector containing only 1's and of length $N$. The relative frequency of the variant $a$ with respect to variant $b$ is simply defined as

$$f_{ab} = \frac{I_a}{I_a + I_b}. \tag{28}$$

We assume that the initial population has reached equilibrium.

We now add a new virus to this population, with index $N + 1$. The new cross-immunity matrix $\tilde{\mathbf{K}}$ is now written as

$$\tilde{\mathbf{K}} = \begin{bmatrix} \mathbf{K} & \vec{b} \\ f & 1 \end{bmatrix}, \tag{29}$$

where $\vec{b}$ and $\vec{f}$ are two vectors of length $N$. This is a general way to write that the backward cross-immunity to variant $a$ caused by an infection with the new variant $N + 1$ is $b_a$. Inversely, the forward cross-immunity to variant $N + 1$ caused by an infection with an old variant $a$ is $f_a$.

This new cross-immunity matrix will of course result in a new equilibrium for the number of infected hosts, given by the vector $\tilde{I}$:

$$\tilde{I} = \frac{\gamma(1 - \delta/\alpha)}{\gamma + \delta} \tilde{\mathbf{K}}^{-1} \vec{1}_{N+1}.w$$

The question we ask here is whether the relative frequency of two variants $1 \leq a, b \leq N$ is changed by the addition of the new variant. In other words, we want to know whether the equality below holds:

$$\frac{I_a}{I_a + I_b} \stackrel{?}{=} \frac{\tilde{I}_a}{\tilde{I}_a + \tilde{I}_b}.$$

Below, we prove this equality under a condition for cross-immunity of the new variant $\vec{b}$ and $\vec{f}$:

$$\vec{b} = b \cdot \vec{1}_N \quad \text{and} \quad \vec{f} = f \cdot \vec{1}_N, \tag{30}$$

where $0 < b, f < 1$ are scalars. This amounts to say that cross-immunity is the same between the new variant $N + 1$ and any old variant $a$, i.e., that the new variant is at an equal antigenic distance from all previous variants.

To prove the equality, we perform the computation $\tilde{\mathbf{K}}^{-1} \vec{1}_{N+1}$. To do that, we make use of the following formula for inverting a block matrix:

$$\begin{bmatrix} A & B \\ C & D \end{bmatrix}^{-1} = \begin{bmatrix} A^{-1} + A^{-1}B\Lambda CA^{-1} & -A^{-1}B\Lambda \\ -\Lambda CA^{-1} & \Lambda \end{bmatrix},$$

where we defined $\Lambda = \left( D - CA^{-1}B \right)^{-1}$. The following identities map to our problem:

$$A = \mathbf{K}, \; B = \vec{b}, \; C = \vec{F}^\mathsf{T}, \; D = 1.$$

We immediately see that $\Lambda = \left( 1 - \vec{f}^\mathsf{T} \mathbf{K}^{-1} \vec{b} \right)^{-1}$ reduces to a scalar that we note $\lambda$ for more clarity. We also define the other scalar value $\mu = \vec{1}_N^\mathsf{T} \mathbf{K}^{-1} \vec{1}_N$. A few manipulations give us the following for $\tilde{\mathbf{K}}^{-1}$:

$$\tilde{\mathbf{K}}^{-1} = \begin{bmatrix} \mathbf{K}^{-1} + \lambda \mathbf{K}^{-1} \vec{b} \cdot \vec{f}^\top \mathbf{K}^{-1} & -\lambda \mathbf{k}^{-1} \vec{b} \\ -\lambda \vec{f}^\top \mathbf{K}^{-1} & \lambda \end{bmatrix}$$

Multiplying this by $\vec{1}_{N+1}$ results in

$$
\begin{aligned}
\tilde{I} &= \tilde{\mathbf{K}}^{-1}\vec{1}_{N+1} \\
&= \left[ \underbrace{\left(\mathbf{K}^{-1} + \lambda\mathbf{K}^{-1}\vec{b}\cdot\vec{f}^{\mathrm{T}}\mathbf{K}^{-1}\right)\vec{1}_N - \lambda\mathbf{K}^{-1}\vec{b}}_{\text{size } N} \,;\, \underbrace{-\lambda\vec{f}^{\mathrm{T}}\mathbf{K}^{-1}\vec{1}_N + \lambda}_{\text{scalar}} \right], \\
&= \left[ I + bf\mu\lambda I - b\lambda I \,;\, \lambda(1 - \mu f) \right] \\
&= \left[ (1 - b\lambda + bf\lambda\mu)I \,;\, \lambda(1 - \mu f) \right]
\end{aligned}
$$

where we used the equalities $\mathbf{K}^{-1}\vec{b} = b\mathbf{K}^{-1}\vec{1}_N = bI$.

This result essentially shows that after adding the new variant, the fraction of hosts infected by the previous variants if simply multiplied by a scalar value $1 - b\lambda(1 - f\mu)$. This implies that the relative frequencies of the original variants are conserved when adding a new one.

## Case with intrinsic fitness effects

In the SIR model of the main text, we assume that the transmission rate $\alpha$ is the same for the different strains. It is also interesting to investigate the case where this transmission rate varies. Here, we study a simple extension of the SIR model without immune groups where there are two variants – mutant and wild-type – with respective transmission rates $\alpha^{wt} = \alpha\phi^{wt}$ and $\alpha^m = \alpha\phi^m$. The quantities $\phi^{wt}, \phi^m \in [\delta/\alpha, \infty]$ can be interpreted as intrinsic fitness values for the two strains. Note that if $\phi^a < \delta/\alpha$, the strain $a$ cannot grow even in a fully susceptible population. The cross-immunity is as usual defined by matrix $\mathbf{K}$ with off-diagonal terms $f$ and $b$.

The equations of motion are now

$$
\dot{S}^a = -\alpha S^a \sum_{b\in\{wt,m\}} \phi^b K^{ab} I^b + \gamma(1 - S^a)
$$

$$
\dot{I}^a = \alpha\phi^a S^a I^a - \delta I^a.
$$

(31)

Computing the equilibrium, we immediately obtain

$$
S^a = \frac{\delta}{\alpha\phi^a}
$$

$$
I = \frac{\gamma}{\delta} \cdot \mathbf{G}^{-1}\vec{h},
$$

(32)

where we have defined the following quantities:

$$
h^a = \frac{\alpha\phi^a - \delta}{\alpha\phi^a}, \quad \mathbf{G} = \begin{pmatrix} 1 & bs \\ fs^{-1} & 1 \end{pmatrix}, \quad s = \frac{\phi^m}{\phi^{wt}}.
$$

(33)

The quantities $h^a$ can be interpreted as a scaled growth rate of each variant given a fully susceptible population, and the matrix $\mathbf{G}$ combines the cross-immunity and the ratio of fitness values $\phi$. Note that it is straightforward to generalize these equations to an arbitrary number of strains: the relevant quantity will be the scaled cross-immunity matrix defined by $G^{ab} = \frac{\phi^b}{\phi^a}K^{ab}$.

Inverting $\mathbf{G}$ and simplifying the equations a bit, we obtain

$$
I^{wt} = \frac{\gamma}{\delta}\frac{\alpha\phi^{wt} - \delta}{\alpha\phi^{wt}} \cdot \frac{1 - b\xi}{1 - bf},
$$

$$
I^m = \frac{\gamma}{\delta}\frac{\alpha\phi^m - \delta}{\alpha\phi^m} \cdot \frac{1 - f\xi^{-1}}{1 - bf},
$$

(34)

where we defined

$$
\xi = \frac{\alpha\phi^m - \delta}{\alpha\phi^{wt} - \delta}.
$$

(35)

Note the interesting structure of **Equations 34**: for each variant, they involve a first term $\frac{\alpha\phi^a - \delta}{\alpha\phi^a}$ that depends only on the intrinsic growth rate of the mutant, and a second $\frac{1 - b\xi}{1 - bf}$ that involves cross-immunity and relative growth rate through $\xi$.

These equilibrium equations give us two conditions for the co-existence of the two variants,

$$b < \xi^{-1} \text{ and } f < \xi, \tag{36}$$

respectively, corresponding to $I^{wt} > 0$ and $I^m > 0$. We mention three interesting cases below.

- If the mutant has an intrinsic fitness disadvantage $\xi < 1$, it will only be able to invade if $f < \xi$. Since $f$ represents the probability that a host becomes immune to the mutant if infected by the wild-type, this means that the immune 'niche' of the mutant must be large enough when compared to $\xi$.
- Invertly, if the mutant is fitter and $\xi > 1$, the mutant is always able to invade. The wild-type only survives if $b < \xi^{-1}$, meaning that the immunity to the wild-type caused by the mutant must be small enough.
- If one considers a situation without total cross-immunity, *i.e.,* $b = f = 1$, the only way a mutant invades is if $\xi > 1$ meaning $\phi^m > \phi^{wt}$, and the result is a full selective sweep.

## Oscillations of the SIR model

The SI model from the main text tends to oscillate while returning to equilibrium. Here, we study this behavior in the simple case of one immune group ($M = 1$) and two viruses (wild-type and variant).

The idea is to linearize the dynamical equations around the equilibrium. This gives us

$$\dot{X} = QX,$$

where $X = [S^{wt}, S^m, I^{wt}, I^m]$ and

$$Q = \begin{pmatrix} -\alpha\gamma/\delta & 0 & -\delta & -\delta b \\ 0 & -\alpha\gamma/\delta & -\delta f & -\delta \\ g_1 & 0 & 0 & 0 \\ 0 & g_2 & 0 & 0 \end{pmatrix}$$

$$g_1 = \frac{\gamma}{\delta}(\alpha - \delta)\frac{1 - b}{1 - bf}$$
$$g_2 = \frac{\gamma}{\delta}(\alpha - \delta)\frac{1 - f}{1 - bf}.$$

To quantify the convergence to equilibrium is the frequency of the oscillations, we need the eigenvalues of matrix $Q$. For low enough $\gamma$, we can prove that the four eigenvalues are

$$\lambda_1 = -\frac{1}{2}\frac{\alpha}{\delta}\gamma \pm i\left(\gamma(\alpha - \delta) - \frac{1}{4}\frac{\alpha^2\gamma^2}{\delta^2}\right)^{1/2}$$

$$\lambda_2 = -\frac{1}{2}\frac{\alpha}{\delta}\gamma \pm i\left(\gamma(\alpha - \delta)\frac{(1 - b)(1 - f)}{1 - bf} - \frac{1}{4}\frac{\alpha^2\gamma^2}{\delta^2}\right)^{1/2}.$$

This is only valid if the terms in the square roots above are positive, which requires $\gamma$ to be small enough. In our setting, we assume $\gamma \ll \alpha, \delta$, so this will always hold.

From the eigenvalues we can compute:

- The rate of convergence to equilibrium: $\alpha\gamma/2\delta$. This means that convergence is slower for a smaller waning rate $\gamma$.
- The two oscillation frequencies that appear:

$$\omega_{high} = \sqrt{\frac{\gamma(\alpha - \delta)}{2\pi}},$$

$$\omega_{low} = \omega_{high}\sqrt{\frac{(1 - b)(1 - f)}{1 - bf}}.$$

Note that since $(1 - b)(1 - f)/(1 - bf) \leq 1$, we always have $\omega_{low} \leq \omega_{high}$. With the values of the main text $\alpha = 3, \delta = 1, \gamma = 5 \cdot 10^{-3}$ (units are inverse of generations), we obtain $\omega_{high} \simeq 0.016$. If one generation is 1 wk, this gives us a period $\omega_{high}^{-1} \simeq 60w$, that is approximately 1 y.

## Link between parameters of the SIR and expiring fitness models

The effective expiring fitness model used in the second part of this work is characterized by the system of differential equations

$$\dot{x} = sx(1 - x) \quad \text{and} \quad \dot{s} = -\nu xs,$$

where $x$ is the frequency of the mutant strain. Here, we try to express the dynamics of the SIR model in this form to find a link between its parameters and the quantities $s$ and $\nu$.

We first focus on the case with two strains and one immune group. The frequency of the mutant is $x = I^m/(I^m + I^{wt})$. Using the logit function $\psi(x) = \log(x/(1 - x))$ and the dynamical equations of the SIR, we find

$$\frac{d}{dt}\psi(x) = \alpha(S^m - S^{wt}), \tag{37}$$

which allows us to define the fitness in the SIR case: $s = \alpha(S^m - S^{wt})$. At the beginning of the invasion, the initial growth rate is readily computed:

$$s(t = 0) = \frac{(1 - f)(\alpha - \delta)}{\delta + f(\alpha - \delta)}\delta, \tag{38}$$

which is the same as the initial growth rate of $I^m$. Note that if $f = 1$, the initial growth rate is 0.

We then compute the time derivative of $s$ early in the invasion, when $I^{wt}, S^{wt}$, and $S^m$ are close to their equilibrium values. In this case, a straightforward calculation gives

$$\dot{s} = -\alpha(I^{wt} + I^m) \cdot \alpha(S^m - bS^{wt}) \cdot x$$
$$\simeq -\alpha(I^{wt} + I^m)xs, \tag{39}$$

where the approximation is valid if $1 - b \ll 1$. This would give an expiry rate of fitness $\nu = \alpha(I^{wt} + I^m)$ in the case of the SIR model.

These results can also be obtained in the case of immune groups. We then have the following expressions for $s$ and $\nu$ at $t \simeq 0$:

$$s = \frac{\alpha}{M}\sum_{i=1}^{M}(S_i^m - S_i^{wt})$$
$$\dot{s} = -\alpha(I^{wt} + I^m)x\sum_{i=1}^{M}(S_i^m - b_iS_i^{wt}) \tag{40}$$
$$= -\alpha(I^{wt} + I^m)xs,$$

where the second expression is again valid if $1 - b_i \ll 1$.

Another question is that of the link between cross-immunity parameters $b$ and $f$ and the distribution of partial sweep sizes $\beta$. The relation between cross-immunity and $\beta$ given by **Equation 6** shows that the distribution of partial sweep size depends on the distribution of both $b$ and $f$. As we do not have a prior on how $b$ and $f$ should be distributed, we explore the case where $1 - f$ and $1 - b$ are exponentially distributed. In other words, we define $\epsilon_f = 1 - f$ and $\epsilon_b = 1 - b$, with the following distributions

$$P(\epsilon_f) \propto e^{-\epsilon_f/\lambda} \quad \text{and} \quad P(\epsilon_b) \propto e^{-\epsilon_b/\mu}.$$

The expression of the partial sweep size becomes $beta = \epsilon_f/(\epsilon_b + \epsilon_f)$. Note that both $\epsilon$ should remain smaller than 1, which is not guaranteed with exponential distributions. However, this is not problematic if μ and $\lambda$ take small enough values.

These assumptions allow us to compute the distribution of $beta$:

$$P(\beta) = \frac{\mu/\lambda}{(\frac{\mu}{\lambda}\beta + (1 - \beta))^2}$$

with support on the interval $0 \leq \beta \leq 1$. **Appendix 1—figure 2** shows the various shapes that $P(\beta)$ then takes for different values of the $\mu/\lambda$ parameter. Note that if $\mu > \lambda$, $b$ tends to be higher than $f$ and $\beta$ is biased towards one. If $\mu = \lambda$, $P(\beta)$ becomes uniform on the $[0, 1]$ range.

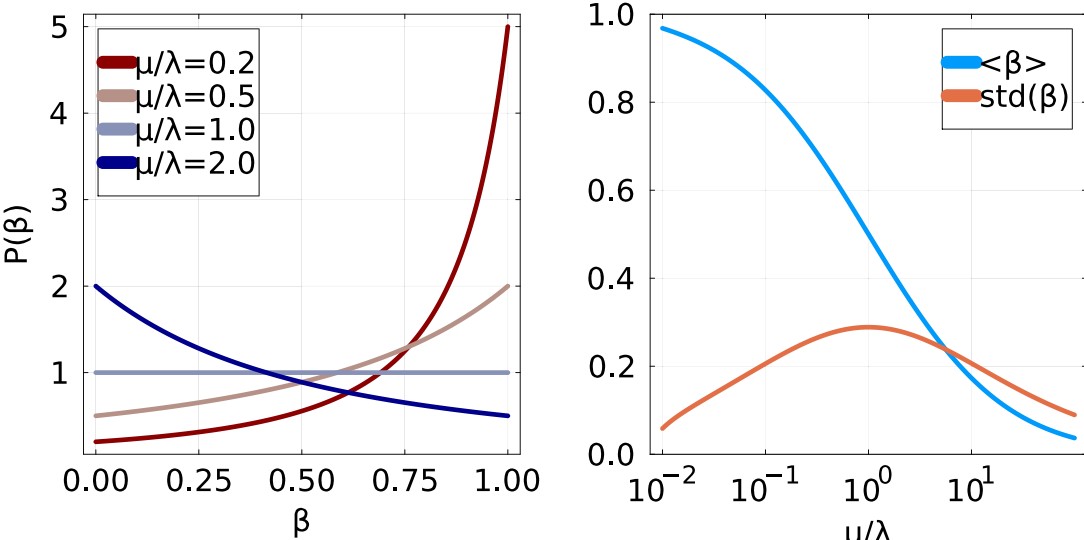

**Appendix 1—figure 2.** Distribution of partial sweep size $\beta$ if $1 - f$ and $1 - b$ are exponentially distributed with respective scales μ and $\lambda$. Left: Probability distribution function $P(\beta)$ for various values of $\mu/\lambda$. Right: Mean and standard deviation of $\beta$ as a function of $\mu/\lambda$.

The exponential distribution away from one of $f$ and $b$ is a reasonable assumption, and allows analytical derivation of $P(\beta)$. Of course, any other distribution of $f$ and $b$ could be considered. For this reason, we choose to use a Beta distribution for $P_\beta$ in the analysis of the main text, as it can accommaodate various shapes.

## Appendix 2

### Expiring fitness model and random walk

#### Clonal interference

In non recombining genomes with a large mutation rate, the appearance of many adaptive mutations in close succession leads to *clonal interference*. In this regime, beneficial mutations present on different background compete for fixation, and the success of a mutation does not depend only on its fitness effect but also on the global state of the population. For this reason, clonal interference causes a decrease in predictability: dynamics are not deterministic but rather depend on the precise structure of which mutation appeared on which background. For instance, a beneficial mutation that increases in frequency can be outcompeted by a fitter one before it has the time to fix, making the extrapolation of frequency trajectories difficult.

We conduct simulations to quantify how much predictability decreases because of clonal interference, based on the ones of the main text. We study a large population of $N = 10^5$ genomes of length $L$, where each genome position $i$ can be in either of two states $\sigma_i \in \{0, 1\}$. Fitness effects $s_i \in \mathbb{R}$ are associated to each position, and the fitness of an individual is $F = \sum_i s_i \sigma_i$. To simulate the adaptation of the population, we proceed in the following way: at a constant rate $\rho$, we pick a position $i$ that is non-polymorphic and set the fitness effect $s_i$ by sampling its magnitude from distribution $P_s$ and choosing its sign in a way that favors mutations (positive if $\sigma_i = 0$ is more frequent, negative otherwise). At the same time, the corresponding mutation ($\sigma_i = 0$ if $s_i < 0$ and inversely) is introduced in the population at a small frequency $\delta f = 0.01$. We choose $P_s$ to be an exponential distribution with scale $s_0 = 0.02$, in agreement with findings on the distribution of fitness effects in *Schiffels et al., 2011*, *Rice et al., 2015*.

There is no fitness decay in this simulation, and the parameters $N$, $s_0$, and $\delta f$ have numerical values such that neutral drift is mostly irrelevant to the dynamics of mutations. For example, without accounting for interference, the probability of fixation of a mutation of effect $s_0$ when it is introduced at frequency $\delta f$ is $p_{fix}(\delta f) \simeq 1 - e^{-40}$. As a consequence, the two parameters governing the dynamics are the scale of fitness effects $s_0$ and the rate of introduction of beneficial mutations $\rho$. The ratio of these two quantities represents the amount of clonal interference: at low $\rho/s_0$, mutations are mostly independent, while at high $\rho/s_0$ they strongly interfere.

We measure the probability of fixation $p_{fix}(x)$ of mutations found in a frequency bin $[x - \delta x, x + \delta x]$ over a long simulation. We only consider mutations with increasing frequency, meaning that their frequency was below $x$ at all times and at some point was measured in the frequency bin. *Figure 4— figure supplement 1* shows $p_{fix}(x)$ as a function of $x$ for different values of $\rho/s_0$. According to intuition, a low clonal interference value leads to easily predictable fixations: whatever the frequency $x$ at which it is observed, a mutation that is increasing in frequency fixes with a very high probability. Increasing clonal interference clearly makes dynamics less predictable and closer to neutrality, with $p_{fix}$ approaching the diagonal line. However, even in a regime of strong interference, e.g., $\rho/s_0 = 32$, deviations from neutrality remain very clear.

#### Expiring fitness effects: Sweep size and probability of overlap

This section gives a few results about the expiring fitness equations from the main text. We rewrite the equations here for reference:

$$\dot{x} = sx(1 - x), \quad \dot{s} = -\nu x s.$$

First, we prove the expression for the amplitude of the partial sweeps. We divide the equation for $\dot{x}$ by the one for $\dot{s}$ to obtain

$$\frac{\mathrm{d}x}{\mathrm{d}s} = \nu^{-1}(x - 1).$$

This immediately gives us

$$x(s) = 1 + \lambda e^{s/\nu}$$

with a constant $\lambda$. At $t = 0$, we have $s = s_0$ and $x = x_0 \ll 1$, while for $t \to \infty$ we have $s = 0$ and $x = \beta$ to be determined. From the $t = 0$ case we obtain $\lambda = (x_0 - 1)e^{-s_0/\nu}$, and from $t \to \infty$ we get $\beta = 1 + \lambda$. Assuming $x_0 \ll 1$, we obtain the result of the main text:

$$\beta = 1 - e^{-s_0/\nu}.$$

We now try to find an expression for the probability that two partial sweeps overlap. First, we try to estimate the time it takes for one partial sweep to complete. While we could not solve the differential equations of the main text analytically, we can give an approximate expression for the time-dependent frequency $x$ during the partial sweep:

$$x(t) \simeq \frac{x_0 \beta}{x_0 + (\beta - x_0)e^{-st}},$$

where $\beta$ is a function of $s$ and $\nu$ and $x_0 = x(t = 0)$. This is simply the expression of a logistic growth starting at $x_0$ and saturating at $\beta$. From there, we compute the time $T_r(s)$ it takes a partial sweep of initial fitness $s$ to reach a frequency $r\beta$ with $x_0\beta^{-1} < r < 1$. We quickly find

$$T_r(s) = s^{-1} \log\left(\frac{\beta}{x_0} \frac{r}{1-r}\right).$$

We now consider that two consecutive partial sweeps of initial fitness values $s_1$ and $s_2$ overlap if the first one is not yet at frequency $r\beta_1$ while the second one is already at $(1-r)\beta_2$. In the figure of the main text, we use $r = 3/4$: an overlap occurs if the first sweep is not yet at 3/4 of its final value while the second one is already at 1/4 of its final value. Thus, for an overlap to occur, we need the time $\tau$ between the two partial sweeps to be smaller than $T_r(s_1) - T_{1-r}(s_2)$. For sweeps happening at rate $\rho$, this has probability $1 - \exp\left(-\rho(T_r(s_1) - T_{1-r}(s_2))\right)$. Since the two sweeps have random initial fitness effects, we find that the overall probability for two consecutive sweeps to overlap is

$$P_r(\text{overlap}) = \int_0^\infty ds_1 ds_2 P_s(s_1) P_s(s_2) \left\{1 - e^{-\rho\left(T_r(s_1) - T_{1-r}(s_2)\right)}\right\}.$$

This integrates over all possible pairs of sweep amplitudes (or initial fitnesses) and weighs them by the probability that the time between the two leads to an overlap. It is this quantity (computed numerically) that is used for the scale of the colorbar in panel E of *Figure 4* of the main text.

## Distribution of partial sweep size $\beta$

This section discusses the distribution of the size of partial sweeps $\beta$ in the context of *Equation 11* of the main text as well as the choice of parameters for panel E of *Figure 4*.

A first interesting case is when fitness effects are exponentially distributed, with parameter $s_0$:

$$P(s) = s_0^{-1} e^{-s/s_0}.$$

This is the distribution we use in most of the population simulations. We compute the corresponding distribution of $\beta$ in a straightforward way:

$$P(\beta < x) = P(s < -\nu \log(1 - x))$$
$$= s_0^{-1} \int_0^{-\nu \log(1-x)} e^{-s/s_0} ds$$
$$= 1 - (1 - x)^{\nu/s_0}.$$

Taking the derivative with respect to $x$, we obtain

$$P(\beta) \propto (1 - \beta)^{\nu/s_0 - 1}.$$

This distribution can accommodate various shape: for $\nu/s_0 > 1$ it peaks at 0, and for $\nu/s_0 < 1$ it peaks at 1. We can also compute the following formula for the moments of $\beta$:

$$\langle \beta \rangle = \frac{s_0}{s_0 + \nu} \quad \text{and} \quad \langle \beta^2 \rangle = \frac{s_0}{s_0 + \nu} \frac{2s_0}{2s_0 + \nu}.$$

When investigating the coalescence time in the main text, we use a different distribution for fitness effects. In this case, we want a finer control over the second moment of $\beta$, and we decide to sample $\beta$ directly using a Beta distribution. The Beta distribution has a support over $[0, 1]$ and can accommodate many different shapes. It is defined by two parameters $a$ and $b$:

$$P(\beta) \propto \beta^{a-1}(1 - \beta)^{b-1}.$$

In our case, it is more practical to parametrize it by its mean $m$ and variance $v$. For given $m$ and $v < m(1 - m)$ we have

$$a = \left( \frac{m(1 - m)}{v} - 1 \right) m, \quad b = \left( \frac{m(1 - m)}{v} - 1 \right)(1 - m)$$

In the case of panel E of **Figure 4** and in order to explore a wide range of distributions, we used three values of $m$, and for each $m$
- a low variance $v = \varepsilon \cdot m^2$ with $\varepsilon = 10^{-5}$
- a high variance $v = m(1 - m)/3$

For a given set of parameters defining a Beta distribution, we decide on the fitness effects by first sampling a $\beta$ for each new adaptive mutation, and then computing $s$ by using **Equation 12** from the main text. For each distribution $P_\beta$, the simulation is performed for 6 values of $\rho \in [0.003, 0.01, 0.018, 0.032, 0.056, 0.1]$.

## Random walk: Monotonous trajectories

Here, we compute the probability that in the random walk defined in the main text, a trajectory starting at $x_0$ converges straight to 0 without ever taking an upward step. While going to 0 requires an infinite amount of downward steps, the probability is still finite since the steps are increasingly likely to go down. For simplicity, we compute this for a fixed $\beta$.

If the random walk always goes down, its position at time step $t$ will be $x_t = (1 - \beta)^t x_0$. Since the probability of going down is $1 - x_t$, the probability of always going down is

$$P_{down} = \prod_{t=0}^{\infty} \left( 1 - (1 - \beta)^t x_0 \right). \tag{41}$$

We simplify this expression by taking the logarithm and assuming that $(1 - \beta)^t \ll 1$ for $t \geq 1$:

$$P_{down} \simeq (1 - x_0) e^{x_0(1 - \beta^{-1})}.$$

Since the random walk is invariant by the change $x \to 1 - x$, we can easily compute the probability of a trajectory always going up, and thus of a monotonous trajectory going straight to either boundary 0 or 1.

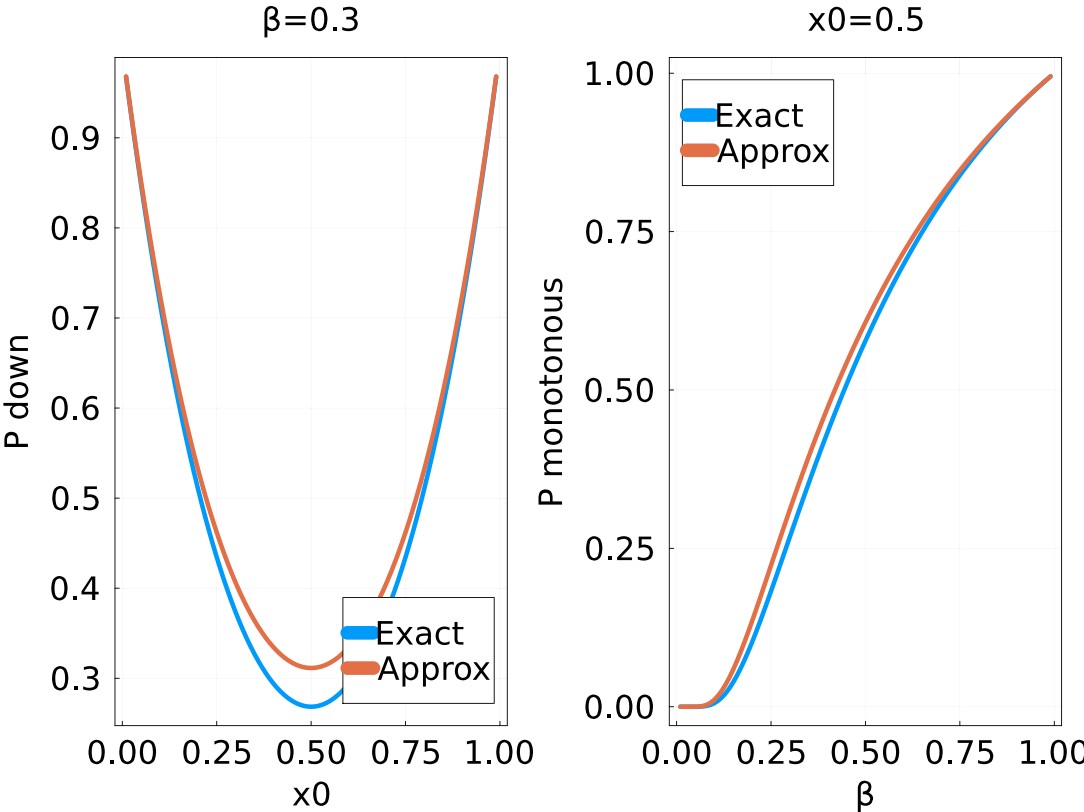

**Appendix 2—figure 1.** Probability of a strictly monotonous trajectory in the random walk of the main text, as a function of $\beta$ (fixed) and the initial value $x_0$. The 'exact' solution is obtained by numerically computing the product in **Equation 41** up to $t = 100$.

The quality of the approximation is quite good, as can be seen from **Appendix 2—figure 1**. The same figure also shows that this probability is relatively high, even for $x_0$ far from either boundary.

## Coalescent

Consider a partial sweep happening between generations $t$ and $t + 1$, with probability $\rho$. One individual $A$ in generation $t$ will then have $\beta N$ children in generation $t + 1$. Any individual in generation $t + 1$ has a probability $\beta$ of having $A$ as a direct ancestor, and a probability $1 - \beta$ of the opposite. If we consider $n$ lineages at generation $t + 1$ and look backward in time, the probability that at least $k$ out of $n$ have $A$ as an ancestor is $\beta^k$. Averaging over $P_\beta$, we find the probability of $k$ specific lineages to have a common ancestor in the previous generation:

$$q_k = \rho \langle \beta^k \rangle.$$

Another useful quantity is the probability $\lambda_n(k)$ that given $n$ lineages, a particular set of exactly $k$ lineages merge one generation back. If a partial sweep of known amplitude $\beta$ took place, this requires the set of $k$ lineages to merge at this generation, with probability $\beta^k$, and that the other $n - k$ do not merge, with probability $(1 - \beta)^{n-k}$.

$$
\begin{aligned}
\lambda_n(k) &= \rho \langle \beta^k (1 - \beta)^{n-k} \rangle, \\
&= \int_0^1 \beta^k (1 - \beta)^{n-k} P(\beta) \mathrm{d}\beta, = \int_0^1 \beta^{k-2} (1 - \beta)^{n-k} \frac{\beta^2 P(\beta)}{\langle \beta^2 \rangle} \mathrm{d}\beta.
\end{aligned}
\tag{42}
$$

This turns out to be the definition of the $\Lambda$-coalescent with $\Lambda(\beta) \propto \beta^2 P(\beta)$ **Schweinsberg, 2000**, **Berestycki, 2009**. The $\Lambda$-coalescent is a general model for genealogies of multiple mergers. We mention two interesting subcases:

- if $P(\beta) = \delta(\beta)$ where $\delta$ is the Dirac distribution with all of the mass at 0, the only possible merge is $k = 2$ and we recover the Kingman coalescent *Berestycki, 2009*. For this reason, we expect our coalescent to approach Kingman's when $\beta \ll 1$, which will be shown explicitly below.
- if $\Lambda(\beta)$ is uniform in $[0, 1]$, meaning $P(\beta) \propto \beta^{-2}$, we obtain the Bolthausen-Sznitman coalescent *Bolthausen and Sznitman, 1998*, *Berestycki, 2009*, which is used to describe the genealogy of populations under strong selection *Bolthausen and Sznitman, 1998*, *Brunet et al., 2007*, *Neher and Hallatschek, 2013*.

Finally, we derive a few more properties of the partial sweep coalescent and show the explicit link to Kingman's when $\beta \ll 1$. Using the $\lambda_n(k)$'s, we can compute the times $T_n$: the time during which exactly $n$ lineages are present in parallel in the genealogy. If there are $n$ lineages present, any coalescence will lower the number of lineages below $n$. The time $T_n$ is thus exponentially distributed with rate $\nu(n)$, where $\nu(n)$ is the total rate of coalescence given $n$ lineages:

$$\nu(n) = \sum_{k=2}^{n} \rho \binom{n}{k} \lambda_n(k).$$

Since we have $\sum_{k=0}^{n} \rho \binom{n}{k} \Lambda_n(k) = \rho(1 - \beta + \beta)^n = \rho$, we finally obtain

$$T_n^{-1} = \rho \left( 1 - \langle (1 - \beta)^n \rangle - n \langle \beta(1 - \beta)^{n-1} \rangle \right). \tag{43}$$

With $n\langle\beta\rangle \ll 1$, we now exactly recover the Kingman coalescent. For simplicity, we assume a constant $\beta$ and expand $T_n$ up to the second order in $n\beta$, to obtain

$$T_n^{-1} = \rho\beta^2 \frac{n(n - 1)}{2} = \frac{n(n - 1)}{2N_e}.$$

These are the times expected for the Kingman coalescent with population size $N_e = 1/\rho\beta^2$.

In the high $n$ limit, we also obtain $T_n \to \rho^{-1}$, since quantities of the type $(1 - \beta)^n$ vanish. This is expected as coalescences only take place when a partial sweep happens, with rate $\rho$. It is another qualitative difference with the Kingman coalescent: since $T_n \geq \rho^{-1}$ for all $n$, one must wait a time $\sim \rho^{-1}$ to observe the first coalescence even in large trees. The shortest branches will thus always be of order $\rho^{-1}$. In contrast, in the Kingman process, the shortest branches vanish when the number of lineages $n$ increases. This difference is clearly visible when looking at terminal branches of trees in *Appendix 2—figure 3*.

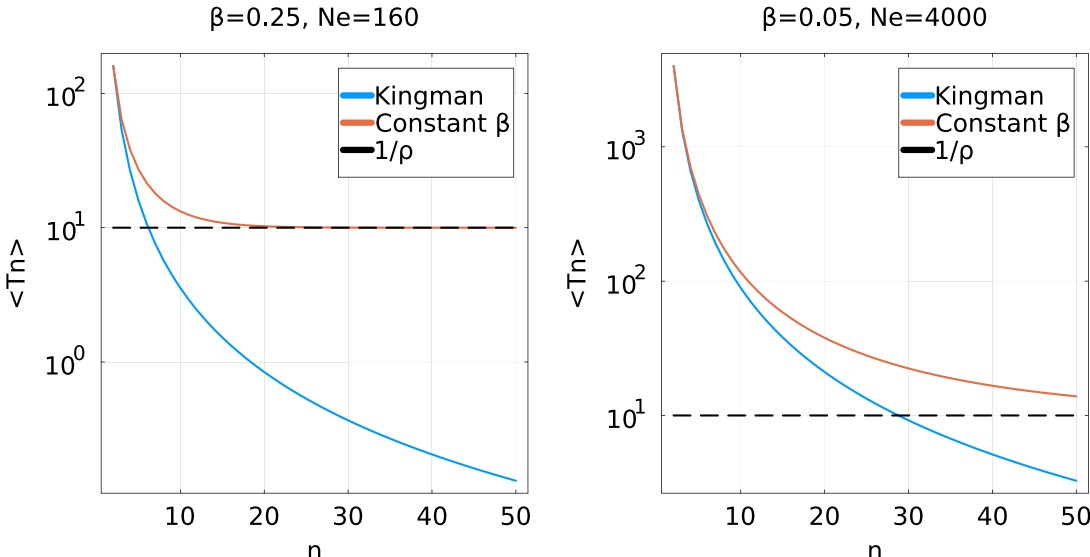

**Appendix 2—figure 2.** Average coalescence times $\langle T_n \rangle$ for a partial sweep coalescent with effective population size $N_e$ and a Kingman coalescent with population size $N_e$. For simplicity, a constant $\beta$ is used: Left: a high value $\beta = 0.25$; Right: a low value $\beta = 0.05$. For low $\beta$, the two coalescent processes are very similar until a high $n$. They considerably differ if $\beta$ is larger. Note that for the partial sweep process, $T_n$ never goes below $\rho^{-1}$.

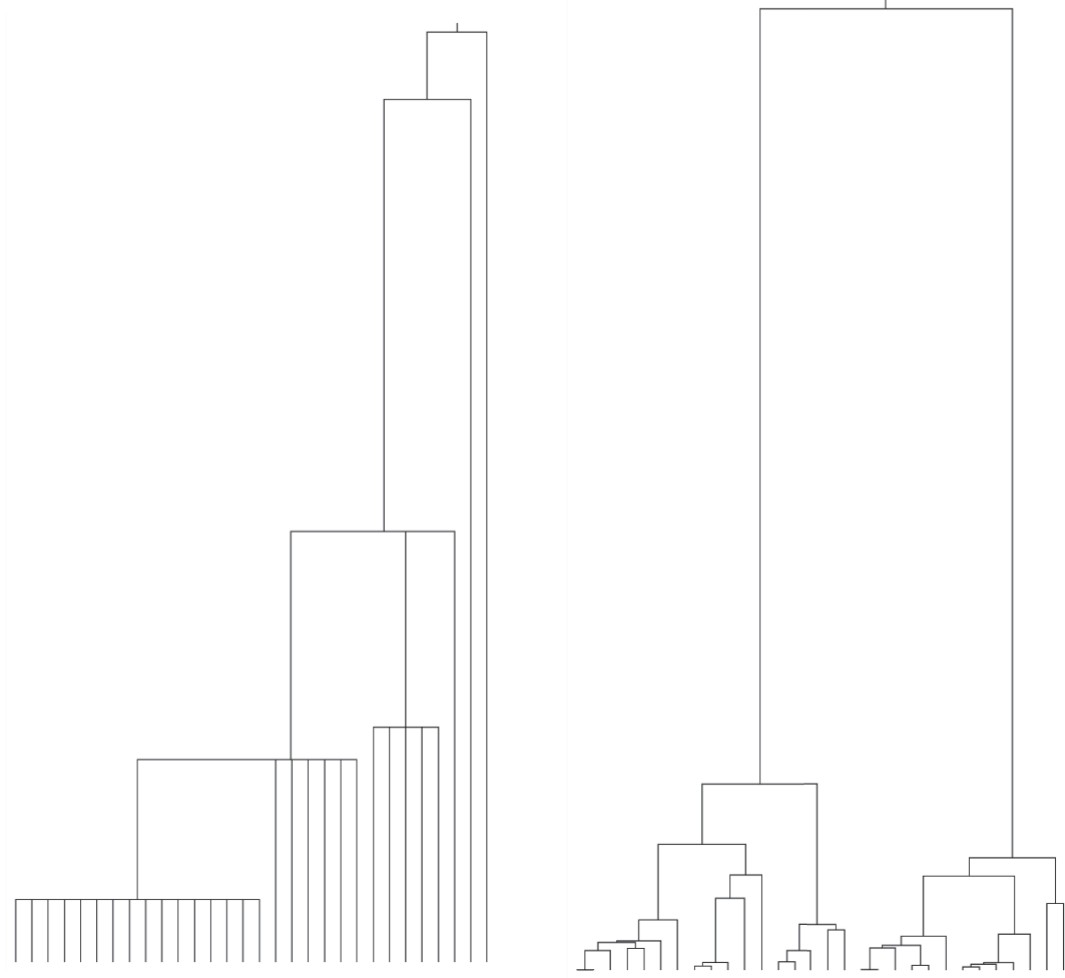

**Appendix 2—figure 3.** Realizations of different coalescence processes for 30 lineages (leaves). Left: Partial sweep coalescent, with constant $\beta = 0.4$ and $\rho = 0.00625$ such that $N_e = (\rho\beta^2)^{-1} = 1\,000$. Right: Kingman coalescent with population size $N = N_e = 1\,000$.

