## [Editor Report · eLife Assessment]

This **important** study provides a new perspective on how human immunity shapes the antigenic evolution of pathogens. By combining theory and simulation the authors make a **compelling** case for the importance of eco-evolutionary interactions in population-level virus-host dynamics, which arise due to coupling between the dynamics of immune memories and viral variants. Although the work does not propose improved data-driven viral forecasting methods, it makes a conceptual contribution that advances the field's understanding of this problem's intrinsic difficulty.

---

## [Referee Report · Reviewer #1 (Public review)]

In this work, the authors study the dynamics of fast-adapting pathogens under immune pressure in a host population with prior immunity. In an immunologically diverse population, an antigenically escaping variant can perform a partial sweep, as opposed to a sweep in a homogeneous population. In a certain parameter regime, the frequency dynamics can be mapped onto a random walk with zero mean, which is reminiscent of neutral dynamics, albeit with differences in higher order moments. Next, they develop a simplified effective model of time dependent selection with expiring fitness advantage, and posit that the resulting partial sweep dynamics could explain the behaviour of influenza trajectories empirically found in earlier work (Barrat-Charlaix et al. Molecular Biology and Evolution, 2021). Finally, the authors put forward an interesting hypothesis: the mode of evolution is connected to the age of a lineage since ingression into the human population. A mode of meandering frequency trajectories and delayed fixation has indeed been observed in one of the long-established subtypes of human influenza, albeit so far only over a limited period from 2013 to 2020. The paper is overall interesting and well-written.

In the revised version, the authors have addressed questions on the role of clonal interference by new simulations in the SI, clarified the connection between the SIR model and vanishing-fitness models, and placed their analysis into the broader context of consumer resource dynamics.

However, the general conclusion, as stated in the abstract, that variant trajectories become unpredictable as a consequence of the SIR dynamics remains somewhat misleading. Two aspects contribute to this problem. (1) The empirical observation of ``quasi-neutrality', i.e. the absence of a net frequency increase inferred as an average of many trajectories at intermediate frequencies, does not imply that individual trajectories are neutral (i.e., fully stochastic and unpredictable) over the time span of observation. Rather, it just says that some have a positive and some have a negative selection coefficient over that time span. (2) As stated by the authors, the observation of average quasi-neutrality is indeed incompatible with the travelling wave model, where initially successful new variants are assumed to retain a fixed, positive selection coefficient from origination to fixation. This observation also limits predictions by extrapolation, where a positive selection coefficient inferred at small frequency is assumed to remain the same at later times and higher frequencies. However, predictions derived from Gog and Grenfell's multi-strain SIR model, as used by several authors, do not make the assumption of fixed selection coefficients and incorporate trajectory-specific, time-dependent expiration effects into their model predictions. This distinction remains blurred throughout the text of the paper.

---

## [Referee Report · Reviewer #3 (Public review)]

In this work the authors present a multi-strain SIR model in which viruses circulate in a heterogeneous population with different groups characterized by different cross-immunity structures. They reformulate the qualitative features of these SIR dynamics as a random walk characterized by new variants saturating at intermediate frequencies. Then they recast their microscopic description to an effective formalism in which viral strains lose fitness independently from one another. They study several features of this process numerically and analytically, such as the average variants frequency, the probability of fixation, and the coalescent time. They compare qualitatively the dynamics of this model to variants dynamics in RNA viruses such as flu and SARS-CoV-2

The idea that vanishing fitness mechanisms that produce partial sweeps may explain important features of flu evolution is very interesting. Its simplicity and potential generality make it a powerful framework. As noted by the authors, this may have important implications for predictability of virus evolution and such a framework may be beneficial when trying to build predictive models for vaccine design. The vanishing fitness model is well analyzed and produces interesting structures in the strains coalescent. Even though the comparison with data is largely qualitative, this formalism would be helpful when developing more accurate microscopic ingredients that could reproduce viral dynamics quantitatively.

This general framework has the potential to be more universal than human RNA viruses, in situations where invading mutants would saturate at intermediate frequencies.

The qualitative connection between the coarse-grained features of these vanishing fitness dynamics and structured SIR processes offers additional intuition relevant to host-pathogens interactions, although as noted by the authors other ecological processes could drive similar evolutionary patterns. The additions in the revised manuscript, substantiating more thoroughly the connection between the SIR and the vanishing fitness description, are important to better appreciate the scope of the work.

---

## [Author Response]

The following is the authors’ response to the original reviews.

**Response to reviewers**

We thank the Editor and the Reviewers for their constructure review. In the light of this feedback, we have made a number of changes and additions to the manuscript, that we think improved the presentation and hopefully address the majority of the concerns by the reviewers.

Main changes:

• We added a new SI section (B1) with a population dynamics simulation in the high clonal interference regime and without expiring fitness (see R1: (1)).

• We added a new SI section (A9) with the derivation of the equilibrium state of our SIR model in the case of 𝑀 immune groups and in the limit 𝜀 → 0 (see R1: (5)).

• The text of the section Abstraction as “expiring” fitness advantage has been modified.

• We added a new SI section (A4) describing the links between parameters of the “expiring fitness” and SIR models.

All three reviewers had concerns about the relation between our SIR model and the “expiring fitness” model, that we hope will be addressed by the last two items listed above. In particular, we would like to underline the following points:

• The goal of our SIR model is to give a mechanistic explanation of partial sweeps using traditional epidemiological models. While ecological models (e.g. consumer resource) can give rise to the same phenomenology, we believe that in the context of host-pathogen interaction it is relevant to explicitely show that SIR models can result in partial sweeps.

• The expiring fitness model is mainly an effective model: it reproduces some qualitative features of the SIR but does not quantitatively match all aspects of the frequency dynamics in SIR models.

• It is possible to link the parameters of the SIR (𝛼,𝛾,𝑏,𝑓) and expiring fitness (𝑠,𝑥,𝜈) models at the beginning of the invasion of the variant (new SI section A4). However, the two models also differ in significant ways (the SIR model can for example oscillate, while the effective model can not). The correspondence of quantities like the initial invasion rate and the ‘expiration rate’ of fitness effects is thus only expected to hold for some time after the emergence of a novel variant.

**Public reviews:**

**Reviewer 1:**
Summary In this work, the authors study the dynamics of fast-adapting pathogens under immune pressure in a host population with prior immunity. In an immunologically diverse population, an antigenically escaping variant can perform a partial sweep, as opposed to a sweep in a homogeneous population. In a certain parameter regime, the frequency dynamics can be mapped onto a random walk with zero mean, which is reminiscent of neutral dynamics, albeit with differences in higher order moments. Next, they develop a simplified effective model of time dependent selection with expiring fitness advantage, and posit that the resulting partial sweep dynamics could explain the behaviour of influenza trajectories empirically found in earlier work (Barrat-Charlaix et al. Molecular Biology and Evolution, 2021). Finally, the authors put forward an interesting hypothesis: the mode of evolution is connected to the age of a lineage since ingression into the human population. A mode of meandering frequency trajectories and delayed fixation has indeed been observed in one of the long-established subtypes of human influenza, albeit so far only over a limited period from 2013 to 2020. The paper is overall interesting and well-written. Some aspects, detailed below, are not yet fully convincing and should be treated in a substantial revision.

We thank the reviewer for their constructive criticism. The deep split in the A/H3N2 HA segment from 2013 to 2020 is indeed the one of the more striking examples of such meandering frequency dynamics in otherwise rapidly adapting populations. But the up and down of H1N1pdm clade 5a.2a.1 in recent years might be a more recent example. We argue that such meandering dynamics might be a common contributor to seasonal influenza dynamics, even if it only spans 3-6 years.

(1) The quasi-neutral behaviour of amino acid changes above a certain frequency (reported in Fig, 3), which is the main overlap between influenza data and the authors’ model, is not a specific property of that model. Rather, it is a generic property of travelling wave models and more broadly, of evolution under clonal interference (Rice et al. Genetics 2015, Schiffels et al. Genetics 2011). The authors should discuss in more detail the relation to this broader class of models with emergent neutrality. Moreover, the authors’ simulations of the model dynamics are performed up to the onset of clonal interference 𝜌/ 𝑠0 = 1 (see Fig. 4). Additional simulations more deeply in the regime of clonal interference (e.g. 𝜌/ 𝑠0 = 5) show more clearly the behaviour in this regime.

We agree with the reviewer that we did not discuss in detail the effects of clonal interference on quasi-neutrality and predictability. As suggested, we conducted additional simulations of our population model in the regime of high clonal interference (𝜌/ 𝑠0 ≫ 1) and without expiring fitness effects. The results are shown in a new section of the supplementary information. These simulations show, as expected, that increasing clonal interference tends to decrease predictability: the fixation probability of an adaptive mutation found at frequency 𝑥 moves closer to 𝑥 as 𝜌 increases. However, even in a case of strong interference 𝜌/ 𝑠0 = 32, 𝑝fix remains significantly different from the neutral expectation. We conclude from this that while it is true that dynamics tend to quasi-neutrality in the case of strong interference, this effect alone is unlikely to explain observations of H3N2 influenza dynamics. In our previous publication (BarratCharlaix et al, MBE, 2021) we have also investigated the effect of epistatic interactions between mutations, along side strong clonal interference. We concluded that, while most of these processes make evolution less predictable and push 𝑝fix towards the diagonal, it is hard to reproduce the empirical observations with realistic parameters. The “expiring fitness” model, however, produces this quite readily.

But there are qualitative differences between quasi-neutrality in traveling wave models and the expiring fitness model. In the traveling wave, a genotype carrying an adaptive mutation is always fitter than if it didn’t carry the mutation. Quasi-neutrality emerges from the accumulation of fitness variation at other loci and the fact that the coalescence time is not much bigger than the inverse selection coefficient of the mutation. In the expiring fitness model, the selective effect of the mutation itself goes away with time. We now discuss the literature on quasi-neutrality and cite Rice et al. 2015 and Schiffels et al. 2011.

In this context, I also note that the modelling results of this paper, in particular the stalling of frequency increase and the decrease in the number of fixations, are very similar to established results obtained from similar dynamical assumptions in the broader context of consumer resource models; see, e.g., Good et al. PNAS 2018. The authors should place their model in this broader context.

We thank the reviewer for pointing out the link between consumer resource models and our work. We further strengthened our discussion of the similarity of the phenomenology to models typically used in ecology and made an effort to highlight the link between consumer-resource models and ours in the introduction and in the part on the SIR model.

(2) The main conceptual problem of this paper is the inference of generic non-predictability from the quasi-neutral behaviour of influenza changes. There is no question that new mutations limit the range of predictions, this problem being most important in lineages with diverse immune groups such as influenza A(H3N2). However, inferring generic non-predictability from quasi-neutrality is logically problematic because predictability refers to individual trajectories, while quasi-neutrality is a property obtained by averaging over many trajectories (Fig. 3). Given an SIR dynamical model for trajectories, as employed here and elsewhere in the literature, the up and down of individual trajectories may be predictable for a while even though allele frequencies do not increase on average. The authors should discuss this point more carefully.

We agree with the reviewer that the deterministic SIR model is of course predictable. Similarly, a partial sweep is predictable. But we argue that expiring fitness makes evolution less predictable in two ways: (i) When a new adaptive mutation emerges and rises in frequency, we typically don’t know how rapidly its fitness effect is ‘expiring’. Thus even if we can measure its instantaneous growth rate accurately, we can’t predict its fate far into the future. (ii) Compared to the situation where fitness effects are not expiring, time to fixation is longer and there are more opportunities for novel mutations to emergence and change the course of the trajectory. We have tried to make this point clearer in the manuscript.

(3) To analyze predictability and population dynamics (section 5), the authors use a Wright-Fisher model with expiring fitness dynamics. While here the two sources of the emerging neutrality are easily tuneable (expiring fitness and clonal interference), the connection of this model to the SIR model needs to be substantiated: what is the starting selection 𝑠0 as a function of the SIR parameters (𝑓,𝑏,𝑀,𝜀), the selection decay 𝜈 = 𝜈(𝑓,𝑏,𝑀,𝜀,𝛾)? This would enable the comparison of the partial sweep timing in both models and corroborate the mapping of the SIR onto the simplified W-F model. In addition, the authors’ point would be strengthened if the SIR partial sweeps in Fig.1 and Fig.2 were obtained for a combination of parameters that results in a realistic timescale of partial sweeps.

We added a new section to the SI (A4) that relates the parameters of the SIR and expiring fitness models. In particular, we compute the initial growth rate 𝑠0 and a proxy for the fitness expiry rate 𝜈 as a function of the SIR parameters 𝛼,𝛾,𝑓,𝑏,𝑀, at the instant where the variant is introduced. The initial growth rate depends primarily on the degree of immune escape 𝑓, while the expiration rate 𝜈 is related to incidence 𝐼wt + 𝐼𝑚. However, as both models have fundamentally different dynamics, these relations are only valid on time scales shorter than potential oscillations of the SIR model. Beyond that, the connection between the models is mostly qualitative: both rely on the fact that growth rate of a strain diminishes when the strain becomes more frequent, and give rise to partial sweeps.

In Figure 1, the time it takes a partial sweep to finish is roughly 100− 200 generations (bottom right panel). If we consider H3N2 influenza and take one generation to be one week, this corresponds to a sweep time of 2 to 4 years, which is slightly slower but roughly in line with observations for selective sweeps. This time is harder to define if oscillatory dynamics takes place (middle right panel), but the time from the introduction of the mutant to the peak frequency is again of about 4 years. The other parameters of the model correspond to a waning time of 200 weeks and immune escape on the order of 20-30% change in susceptibility.

**Reviewer 2:**
SummaryThis work addresses a puzzling finding in the viral forecasting literature: high-frequency viral variants evince signatures of neutral dynamics, despite strong evidence for adaptive antigenic evolution. The authors explicitly model interactions between the dynamics of viral adaptations and of the environment of host immune memory, making a solid theoretical and simulation-based case for the essential role of host-pathogen eco-evolutionary dynamics. While the work does not directly address improved data-driven viral forecasting, it makes a valuable conceptual contribution to the key dynamical ingredients (and perhaps intrinsic limitations) of such efforts.StrengthsThis paper follows up on previous work from these authors and others concerning the problem of predicting future viral variant frequency from variant trajectory (or phylogenetic tree) data, and a model of evolving fitness. This is a problem of high impact: if such predictions are reliable, they empower vaccine design and immunization strategies. A key feature of this previous work is a “traveling fitness wave” picture, in which absolute fitnesses of genotypes degrade at a fixed rate due to an advancing external field, or “degradation of the environment”. The authors have contributed to these modeling efforts, as well as to work that critically evaluates fitness prediction (references 11 and 12). A key point of that prior work was the finding that fitness metrics performed no better than a baseline neutral model estimate (Hamming distance to a consensus nucleotide sequence). Indeed, the apparent good performance of their well-adopted “local branching index” (LBI) was found to be an artifact of its tendency to function as a proxy for the neutral predictor. A commendable strength of this line of work is the scrutiny and critique the authors apply to their own previous projects. The current manuscript follows with a theory and simulation treatment of model elaborations that may explain previous difficulties, as well as point to the intrinsic hardness of the viral forecasting inference problem.This work abandons the mathematical expedience of traveling fitness waves in favor of explicitly coupled eco-evolutionary dynamics. The authors develop a multi-compartment susceptible/infected model of the host population, with variant cross-immunity parameters, immune waning, and infectious contact among compartments, alongside the viral growth dynamics. Studying the invasion of adaptive variants in this setting, they discover dynamics that differ qualitatively from the fitness wave setting: instead of a succession of adaptive fixations, invading variants have a characteristic “expiring fitness”: as the immune memories of the host population reconfigure in response to an adaptive variant, the fitness advantage transitions to quasi-neutral behavior. Although their minimal model is not designed for inference, the authors have shown how an elaboration of host immunity dynamics can reproduce a transition to neutral dynamics. This is a valuable contribution that clarifies previously puzzling findings and may facilitate future elaborations for fitness inference methods.The authors provide open access to their modeling and simulation code, facilitating future applications of their ideas or critiques of their conclusions.

We thank the reviewer for their summary, assessement, and constructive critique.

(1) The current modeling work does not make direct contact with data. I was hoping to see a more direct application of the model to a data-driven prediction problem. In the end, although the results are compelling as is, this disconnect leaves me wondering if the proposed model captures the phenomena in detail, beyond the qualitative phenomenology of expiring fitness. I would imagine that some data is available about cross-immunity between strains of influenza and sarscov2, so hopefully some validation of these mechanisms would be possible.

We agree with the reviewer that quantitatively confronting our model with data would be very interesting. Unfortunately, most available serological data for influenza and SARS-CoV-2 is obtained using post-infection sera from previoulsy naive animal models. To test our model, we would require human serology data, ideally demographically resolved, and a way to link serology to transmission dynamics. Furthermore, our model is mostly an explanation for qualitative features of variant dynamics and their apparent lack of predictability. We therefore considered that quantitative validation using data is out of scope of this work.

(2) After developing the SIR model, the authors introduce an effective “expiring fitness” model that avoids the oscillatory behavior of the SIR model. I hoped this could be motivated more directly, perhaps as a limit of the SIR model with many immune groups. As is, the expiring fitness model seems to lose the eco-evolutionary interpretability of the SIR model, retreating to a more phenomenological approach. In particular, it’s not clear how the fitness decay parameter 𝜈 and the initial fitness advantage 𝑠0 relate to the key ecological parameters: the strain cross-immunity and immune group interaction matrices.

The expiring fitness model emerges as a limiting case, at least qualitatively, of the SIR model when growth rate of the new variant is small compared to the waning rate and the SIR model does not oscillate. This can be readily achieved by many immune groups, which reconciles the large effect of many escape mutations and the lack of oscillation by confining the escape to some fraction of the population. Beyond that, the expiring fitness model is mainly an effective model that allows us to study the consequences of partial sweeps on predictability on long timescales. As stated in the “Main changes” section at the start of this reply, we added an SI section which links parameters of the two models. However, we underline the fact that beyond the phenomenon of partial sweeps, the dynamics of the two are different.

**Reviewer 3:**
SummaryIn this work the authors start presenting a multi-strain SIR model in which viruses circulate in an heterogeneous population with different groups characterized by different cross-immunity structures. They argue that this model can be reformulated as a random walk characterized by new variants saturating at intermediate frequencies. Then they recast their microscopic description to an effective formalism in which viral strains lose fitness independently from one another. They study several features of this process numerically and analytically, such as the average variants frequency, the probability of fixation, and the coalescent time. They compare qualitatively the dynamics of this model to variants dynamics in RNA viruses such as flu and SARS-CoV-2.StrengthsThe idea that a vanishing fitness mechanisms that produce partial sweeps may explain important features of flu evolution is very interesting. Its simplicity and potential generality make it a powerful framework. As noted by the authors, this may have important implications for predictability of virus evolution and such a framework may be beneficial when trying to build predictive models for vaccine design. The vanishing fitness model is well analyzed and produces interesting structures in the strains coalescent. Even though the comparison with data is largely qualitative, this formalism would be helpful when developing more accurate microscopic ingredients that could reproduce viral dynamics quantitatively. This general framework has a potential to be more universal than human RNA viruses, in situations where invading mutants would saturate at intermediate frequencies.

We thank the reviewer for their positive remarks and constructive criticism below.

WeaknessesThe authors build the narrative around a multi-strain SIR model in which viruses circulate in an heterogeneous population, but the connection of this model to the rest of the paper is not well supported by the analysis. When presenting the random walk coarse-grained description in section 3 of the Results, there is no quantitative relation between the random walk ingredients importantly 𝑃(𝛽) - and the SIR model, just a qualitative reasoning that strains would initially grow exponentially and saturate at intermediate frequencies. So essentially any other microscopic description with these two features would give rise to the same random walk.

As also highlighted in the response to other reviewers, we now discuss how the parameter of the SIR model are related to the initial growth rate and the ‘expiration’ rate of the effective model. While the phenomenology of the SIR model is of course richer, this correspondence describes its overdamped limit qualitatively well.

Currently it’s unclear whether the specific choices for population heterogeneity and cross-immunity structure in the SIR model matter for the main results of the paper. In section 2, it seems that the main effect of these ingredients are reduced oscillations in variants frequencies and a rescaled initial growth rate. But ultimately a homogeneous population would also produce steady state coexistence between strains, and oscillation amplitude likely depends on parameters choices. Thus a homogeneous population may lead to a similar coarse-grained random walk.

The reviewer is correct that the primary effects of using many immune groups is to slow down the increase of novel variant, which in turn dampens the oscillations. Having multiple immune groups widens the parameter space in which partial sweeps without dramatic oscillations are observed. For slow sweeps, similar dymamics are observed in a homogeneous population.

Similarly, it’s unclear how the SIR model relates to the vanishing fitness framework, other than on a qualitative level given by the fact that both descriptions produce variants saturating at intermediate frequencies. Other microscopic ingredients may lead to a similar description, yet with quantitative differences.

Both of these points were also raised by other reviewers and we agree that it is worth discussing them at greater length. We now discuss how the parameters of the ‘expiring fitness’ model relate to those of the SIR. We also discuss how other models such as ecological models give rise to similar coarse grained models.

At the same time, from the current analysis the reader cannot appreciate the impact of such a mean field approximation where strains lose fitness independently from one another, and under what conditions such assumption may be valid.

In the SIR model, the rate at which strains lose fitness does depend on the precise state of the host population through the quantities 𝑆𝑚 and 𝑆wt , which is apparent in equation (A27) of the new SI section. The fact that a new variant shifts the equilibrium frequencies of previous strains in a proportional way is valid if the “antigenic space” is of very high dimensions, as explained in section Change in frequency when adding subsequent strains of the SI. It would indeed be interesting to explore relaxations of this assumption by considering a larger class of cross immunity matrices 𝐾. However, in the expiring fitness model, the fact that strains lose fitness independently from each ohter is a necessary simplification.

In summary, the central and most thoroughly supported results in this paper refer to a vanishing fitness model for human RNA viruses. The current narrative, built around the SIR model as a general work on host-pathogen eco-evolution in the abstract, introduction, discussion and even title, does not seem to match the key results and may mislead readers. The SIR description rather seems one of the several possible models, featuring a negative frequency dependent selection, that would produce coarse-grained dynamics qualitatively similar to the vanishing fitness description analyzed here.

We have revised the text throughout to make the connections between the different parts of the manuscript, in particular the SIR model and the expiring fitness model, clearer. We agree that the phenomenology of the expiring fitness model is more general than the case of human RNA viruses described by the SIR model, but we think this generality is an attractive feature of the coarse-graining, not a shortcoming. Indeed, other settings with negative frequency dependent selection or eco-systems that adapt on appropriate time scale generate similar dynamics.

**Recommendations for the authors:**

**Reviewer 1:**

(4) Line 74: what does fitness mean?

Many population dynamics models, including ones used for viral forecasting, attach a scalar fitness to each strain. The growth rate of each strain is then computed by substracting the average population fitness to the strain’s fitness. In this sentence, fitness is intended in this way.

(5) Fig. 1: The equilibrium frequency in the middle and bottom rows is hardly smaller than the equilibrium frequency in the top row for one immune group. This is surprising since for M=10, the variant escapes in only 1/10th of the population, which naively should impact the equilibrium frequency more strongly. Could the authors comment on this?

This is indeed non-trivial, and a hand-waving argument can be made by considering the extreme case 𝜀 = 0. The variant is then completely neutral for the immune groups 𝑖 > 1, and would be at equilibrium at any frequency in these immune groups. Its equilibrium frequency is then only determined by group 1, which is the only one breaking degeneracy. For 𝜀 > 0 but small, we naturally expect a small deviation from the 𝜀 = 0 case and thus 𝛽 should only change slightly.

A more rigorous argument with a mathematical proof in the case 𝜀 = 0 is now given in section A4 of the supplementary information.

(6) Fig. 1: In the caption, it is stated that the simulations are performed with 𝜀 = 0.99. Is this a typo? It seems that it should be 𝜀 = 0.01, as in and just below equation (7).

This was indeed a typo. It is now fixed.

(7) Fig. 3: The data analysis should be improved. In order to link the average frequency trajectories to standard population genetics of conditional fixation probabilities, the focal time should always be the time where the trajectory crosses the threshold frequency for the first time. Plotting some trajectories from a later time onwards, on their downward path destined to loss, introduces a systematic bias towards negative clonal interference (for these trajectories, the time between the first and the second crossing of the threshold frequency is simply omitted). The focal time of first crossing of the threshold frequency can easily be obtained, e.g., by linear interpolation of the trajectory between subsequent time points of frequency evalution. In light of the modified procedure, the statements on the on the inertia of the trajectories after crossing 𝑥⋆ (line 356) should be re-examined.

The way we process the data is already in line with the suggestions of the reviewer. In particular, we use as focal time the first time at which a trajectory is found in the threshold frequency bin. Trajectories that are never seen in the bin because of limited time-resolution are simply ignored.

In Fig. 3, there are no trajectories that are on their downward path at the focal time and when crossing the threshold frequency. Our other work on predictability of flu Barrat-Charlaix et. al. (2021) has a similar figure, which maybe created confusion.

(8) Fig. 4: authors write 𝛼/ 𝑠0 in the figure, but should be 𝜈/ 𝑠0.

Fixed.

(9) Line 420: authors refer to the blue curve in panel B as the case with strong interference. However, strong interference is for higher 𝜌/ 𝑠0, that is panel D (see point 1).

Fixed.

(10) Line 477: typo “there will a variety of mutations”.

Fixed.

**Reviewer 2:**
Should 𝛼 be 𝜈 in Figure 4 legends?

Thank you very much for spotting this error. We fixed it.

Equations 4-5 could be further simplified.

We factorised the 𝐼 term in equation 4. In equation 5, we prefered to keep the 1− 𝛿/ 𝛼 term as this quantity appears in different calculations concerning the model. For instance, 𝑆 = 𝛿/ 𝛼 at equilibrium.

The sentence before equation 8 references 𝑃𝛽(𝛽), but this wasn’t previously introduced.

We now introduce 𝑃𝑏𝜂 at the beginning of the section Ultimate fate of the variant.

In the last paragraph of page 12, “monotonously” maybe should be “monotonically”.

Fixed.

For the supplement section B, you might want a more descriptive title than “other”.

We renamed this section to Expiring fitness model and random walk.

**Reviewer 3:**
To expand on my previous comments, my main concerns regard the connection of section 2 and the SIR model with the rest of the paper.In the first paragraph of page 9 the authors argue that a stochastic version of the SIR model would lead to different fixation dynamics in homogeneous vs heterogeneous populations due to the oscillations. This paragraph is quite speculative, some numerical simulations would be necessary to quantitatively address to what extent these two scenarios actually differ in a stochastic setting, and how that depends on parameters.Likewise, the connection between the SIR model, the random walk coarse-grained description and the vanishing fitness model can be investigated through numerical simulations of a stochastic SIR given the chosen population and cross-immunity structures with i.e. 10-20 strains. This would allow for a direct comparison of individual strain dynamics rather than the frequency averages, as well as other scalar properties such as higher moments, coalescent, and fixation probability once reaching a given frequency. It would also be possible to characterize numerically the SIR P(beta) bridging the gap with the random walk description. It’s not obvious to me that the SIR P(beta) would not depend on the population size in the presence of birth-death stochasticity, potentially changing the moments scalings. I appreciate that such simulations may be computationally expensive, but similar numerical studies have been performed in previous phylodynamics works so it shouldn’t be out of reach.An alternative, the authors should consider re-centering the narrative directly on the random walk of the vanishing fitness model, mentioning the SIR more briefly as a possible qualitative way to get there. Either way the authors should comment on other ways in which this coarse-grained dynamics could arise.In the vanishing fitness model, where variants fitnesses are independent, is an infinite dimensional antigenic space implicitly assumed? If that’s the case, it should be explained in the main text.

A long simulation of the SIR model would indeed be interesting, but is numerically demanding and our current simulation framework doesn’t scale well for many strains and susceptibilities. We thus refrained from adding extensive simulations.

In Figure 2B of the main text, the simulation with 7 strains illustrates the qualitative match between the expiring fitness and the SIR model. However, it is clearly not long enough to discuss statistical properties of the corresponding random walk. Furthermore, we do not expect the individual strain dynamics of the SIR and expiring fitness models to match. The latter depends on few parameters (𝛼, 𝑠0), while the former depends on the full state of the host population and of the previous variants.

In the sectin linking the parameters of the two models, we now discuss the distribution 𝑃(𝛽) of the SIR model for two strains and a specific choice of distribution for the cross immunity 𝑏 and 𝑓.

Minor comments:There is some back and forth in the writing. For instance, when introducing the model, 𝐶𝑖𝑗 is first defined as 1/ 𝑀, then a few paragraphs later the authors introduce that in another limit 𝐶𝑖𝑖 is just much higher than any 𝐶𝑖𝑗, and finally they specify that the former is the fast mixing scenario.Another example is in section 2, in the first paragraph they put forward that heterogeneity and crossimmunity have different impacts on the dynamics, but the meaning attributed to these different ingredients becomes clear only a while later after the homogeneous population analysis. Uniforming the writing would make it easier for the reader to follow the authors’ train of thought.

We removed the paragraph below Equation (1) mentioning the 𝐶𝑖𝑗 = 1/ 𝑀 case, which we hope will linearize the writing.

When mentioning geographical structure, why would geography affect how immunity sees pairs of viral strains (differences in 𝐾)?

Geographic structure could influence cross-immunity because of exposure histories of hosts. For instance in the case of influenza, different geographical regions do not have the same dominating strains in each season, and hosts from different regions may thus build up different immunity.

In the current narrative there are some speculations about non-scalar fitness, especially in section 2. The heterogeneity in this section does not seem so strong to produce a disordered landscape that defies the notion of scalar fitness in the same way some complex ecological systems do. A more parsimonious explanation for the coexistence dynamics observed here may be a negative frequency dependent selection.

Our language here was not very precise and we agree that the phenomenology we describe is related to that of frequency dependent selection (mediated by via immunity of the host population that integrates past frequencies). Traveling wave models typically use fitness function that are independent of the population distribution and only account for the evolution via an increasing average fitness. We have made discussion more accurate by stating that we consider a case where fitness depends explicitly on present and past population composition, which includes the case of negative frequency dependent selection.

I don’t understand the comparison with genetic drift (typo here, draft) in the last paragraph of section 3 given that there is no stochasticity in growth death dynamics.

We compare the random walk to genetic drift because of the expression of the second moment of the step size. The genetic draft has the same functional form. If one defines the effective population size as in the text, the drift due to random sampling of alleles (neutral drift) and the changes in strain frequency in our model have the same first and second moments. The stochasticity here does not come from the dynamics, which are indeed deterministic, but from the appearance of new mutations (variants) on backgrounds that are randomly sampled in the population. This latter property is shared with genetic draft.

In the vanishing fitness model, I think the reader would benefit from having 𝑃(𝑠) in the main text, and it should be made more clear what simulations assume what different choice of 𝑃(𝑠).

We added the expression of 𝑃(𝑠) in the main text. Simulations use the value 𝑠0 = 0.03, which we added in the caption of Figure 4.

When comparing the model and data, is the point that COVID is not reproduced due to clonal interference? It seems from the plot that flu has clonal interference as well though. Why is that negligible?

A similar point has been raised by the first reviewer (see R1-(1)). Clonal interference is not negligible, but we find it to be insufficient to explain the observations made for H3N2 influenza, namely the lack of inertia of frequency trajectories or the probability of fixation. This is shown in the new section (B1) of the SI. Both SARS-CoV-2 and H3N2 influenza experience clonal interference, but the former is more predictable than the latter. Our point is that expiring fitness effects should be stronger in influenza because of the higher immune heterogeneity of the host population, making it less predictable than SARS-CoV-2.

Does the fixation probability as a function of frequency threshold match the flu data for some parameters sets?

For H3N2 influenza, the fixation probability is found to be equal to the threshold frequency (see Barrat-Charlaix MBE 2021, also indirectly visible from Fig. 3). In Figure 4, we obtain that either a high expiry rate or intermediate expiry rates and clonal interference regimes match this observation.

It would be instructive to see examples of the individual variant dynamics of the vanishing fitness model compared to the presented data.

We added an extra SI figure (S7) showing 10 randomly selected trajectories of individual variants in the case of H3N2/HA influenza and for the expiring fitness model with different parameter choices.

Figure 4E has no colorbar label. The reader shouldn’t have to look for what that means in the bottom of the SIs. In panels A and B the label should be 𝜈, not 𝛼. Same thing in most equations of page 42.

We added the colorbar label to the figure and also updated the caption: a darker color corresponds to a higher probability of sweeps to overlap. We fixed the 𝜈 – 𝛼 confusion in the SI and in the caption of the figure.